



# Induced seismicity risk analysis of the hydraulic stimulation of a geothermal well on Geldinganes, Iceland

Marco Broccardo[1,4], Arnaud Mignan[2,3], Francesco Grigoli[1], Dimitrios Karvounis[1], Antonio Pio Rinaldi[1,2], Laurentiu Danciu[1], Hannes Hofmann[5], Claus Milkereit[5], Torsten Dahm[5], Günter Zimmermann[5], Vala Hjörleifsdóttir[6], Stefan Wiemer[1]

[1] Swiss Seismological Service, ETH Zürich, Switzerland
[2] Institute of Geophysics, ETH Zürich, Switzerland
[3] Institute of Risk Analysis, Prediction and Management, Academy for Advanced Interdisciplinary Studies, Southern University of Science and Technology, Shenzhen, China
[4] Institute of Structural Engineering, ETH Zürich, Switzerland
[5] Helmholtz Centre Potsdam GFZ German Research Centre for Geosciences, Potsdam, Germany
[6] Orkuveita Reykjavíkur/Reykjavík Energy, Iceland

*Correspondence to*: Marco Broccardo (bromarco@ethz.ch)

**Abstract.** The rapid increase in energy demand in the city of Reykjavik has posed the need for an additional supply of deep geothermal energy. The deep hydraulic (re-)stimulation of well RV-43 on the peninsula of Geldinganes (north of Reykjavik) is an essential component of the plan implemented by Reykjavik Energy to meet this energy target. Hydraulic stimulation is often associated with fluid-induced seismicity, most of which is not felt on the surface, but which, in rare cases, can cause nuisance to the population and even damage to the nearby building stock. This study presents a first of its kind pre-drilling probabilistic induced-seismic hazard and risk analysis for the site of interest. Specifically, we provide probabilistic estimates of peak ground acceleration, European microseismicity intensity, probability of light damage (damage risk), and individual risk. The results of the risk assessment indicate that the individual risk within a radius of 2 km around the injection point is below 0.1 micromorts, and damage risk is below $10^{-2}$, for the total duration of the project. However, these results are affected by several orders of magnitude of variability due to the deep uncertainties present at all levels of the analysis, indicating a critical need in updating this a-priory risk assessment with in-situ data collected during the stimulation.

## 1 Introduction

The city of Reykjavik, the capital and center of population of Iceland, meets 99.9% of its district heating demand by geothermal energy (Gunnlaugsson et al., 2000). However, the growing population and the booming number of tourists is pushing the



current supply of energy to its limit, since no new low temperature wells have been drilled since 2001. In particular, additional sources of low temperature heat need to be accessed to ensure a reliable heat provision for the city center. Therefore, there is an urgent need to increase the current capacity by drilling new low temperature wells and stimulating older inactive wells.

One potential area for new low temperature geothermal field developments is Geldinganes. Geldinganes is a peninsula within the city limits of Reykjavik (Figure 1). The exceptional geothermal gradient in this area triggered the drilling of a well (RV-43) in 2001 after a gabbro body was identified as potential heat source and drilling target for this deviated well. Despite the required temperatures being reached, the flow rates were insufficient for economic production. At present, Reykjavik Energy (Orkuveita Reykjavíkur, OR) re-assessed this field for development of geothermal energy with new production wells. To additionally enhance the production, it is foreseen to hydraulically re-stimulate well RV-43 in order to improve its productivity to economical levels. In particular, a three-staged cyclic pulse stimulation is planned that will last for (circa) 12 days. The stimulation is expected to enhance productivity in three pre-existing fracture zones penetrated by RV-43 and isolated with straddle packers. Packers technology, commonly used in the oil and gas industry, is expected to be employed for the upcoming stimulation. This will allow isolating selected narrow zones of the well and thus injecting exclusively through them.

Like all energy technologies, the exploitation of deep geothermal energy is not risk-free. Therefore, an essential part of the implementation and licensing is a quantitative risk assessment comparable to existing regulations for Health, Safety, Environment (HSE) procedures. This analysis allows balancing the (perceived and real) risks against the (perceived and real) benefits. Over the last decade, induced seismicity has emerged as one of the risks—often the most dominant one—to be faced (Giardini, 2009, Grigoli et al., 2017) in implementing industrial underground technologies (e.g., geothermal energy exploitation, water impoundment, $CO_2$-sequestration and natural gas storage operations, non-conventional hydrocarbon production, etc.). These activities can alter the stress field of the shallow Earth's crust by pore pressure changes, or volume and/or mass changes inducing or triggering seismicity (Ellsworth, 2013; Giardini, 2009; Mignan, 2016). Such earthquakes pose a nuisance or even a danger to the local population and can strongly undermine the societal acceptance of a project (Trutnevyete and Wiemer, 2017, Grigoli et al. 2017, Hirschberg et al., 2015). The recent M5.5 Pohang earthquake (Grigoli et al., 2018; Kim et al., 2018) is an extreme example of a triggered earthquake related to geothermal activities that had a combined economic impact of $300M as well as more than 135 injuries (Lee et al., 2019). In particular, fluid injection or extraction in tectonically active zones carries a risk of inducing a seismic event of a significant magnitude (Grigoli et al. 2017), and deep geothermal projects are a primary example. Adding an additional concern, deep geothermal projects in Europe—and the Geldinganes stimulation is no exception—are often located close to consumers, thus in densely urbanized areas with historical and hence vulnerable buildings and infrastructures. In these contexts, the problem of assessing and managing induced seismicity is especially important (Mignan et al., 2015; 2019a; 2019b). It is also a well-known fact that societal acceptance of



induced seismicity has substantially decreased in some countries in the past decade, a result of failures discussed widely in the media and an overall change in risk perception.

Despite the large body of research conducted over the past decades by numerous research groups, the physical,
65    chemical and hydro-mechanical mechanisms governing induced seismicity are far from being fully understood, posing clear limits to the risk assessment and management strategies (Yeck et al., 2017; Trutnevyete and Wiemer, 2017; Grigoli et al., 2017; Mignan et al., 2019a; 2019b). The limitations to forecast induced seismicity are, on the one hand, the non-uniqueness on the physical framework for modelling, and on the other hand, even more important, the large uncertainties on the boundary conditions needed for forecasting (e.g., where are faults, what are their sizes and stressing state, what is the permeability
70    distribution of the reservoir, etc.). It follows that any risk assessment and management strategy must capture the existing uncertainties and lack of knowledge, requiring a probabilistic approach that explicitly considers both epistemic uncertainties and aleatory variability. It also implies that in order to reduce uncertainties, the risk assessment should be updated as soon as new data becomes available during the drilling and stimulation phase (e.g., Broccardo et al., 2017a).

Despite these challenges, geothermal energy is a highly important renewable energy resource with a low carbon
75    footprint. It has been successfully operated in many areas for decades, and Iceland is a prime example for economically successful and widely accepted use of deep geothermal energy. Past geothermal projects have been successfully managed with classical traffic light approaches (Majer et al. 2007; Bommer et al. 2006; Kwiatek et al. 2019) and simplified a-priori risk assessments. However, we consider it important for the future development of geothermal energy near urbanised areas to move beyond the existing state of the technology and develop and implement a robust, quantitative, and coherent risk management
80    framework during all stages of a project.

Within this context, this study represents, to the best of our knowledge, the first probabilistic seismic risk study prior to a deep geothermal project in Iceland (and one of the very few worldwide). We have attempted to combine all available risk-related information on the upcoming stimulation of the RV-43 well in Geldinganes into one quantitative and risk-based assessment. This a-priori study, then, represents the basis for risk updating once the project has started and in-situ real-time
85    data become available. This procedure ideally enables a dynamic risk management solution that will also help to ensure public acceptance, and thus contribute to the continued successful use of deep geothermal energy resources in Iceland and beyond.

Our paper is structured as follow: Section 2 describes the site, the geological conditions, and the planned field operations; Section 3 introduces the probabilistic fluid-induced seismic hazard assessment and Section 4 the probabilistic fluid-induced seismic risk assessment; Section 5 discusses the hazard and risk results as well as known limitations.



## 2 Site description, geological conditions, and planned operations

### 2.1 Site description

Well RV-43 is located on the Geldinganes peninsula in the northeastern part of the city of Reykjavik Figure 1. OR is the main supplier of heat in Reykjavik and has drilled several wells on Geldinganes. It aims to produce hot water from RV-43 to be directly utilized for heating purposes and to meet the increasing energy needs of Reykjavik.

RV-43 was drilled in 2001; it is 1832 m long, where the last 1130 m are uncased (8½ inches open hole). The well is deviated towards N20°E (on average) and it reaches ~1550 m true vertical depth (TVD). The well is oriented towards the northeast of Geldinganes, an area with exceptionally high geothermal gradients that is closer than the rest of the Geldinganes' wells to the extinct central volcanic system north of Reykjavik and to a possible fault zone (Steingrimsson et al., 2001). Both temperature logs and magnetic measurements are supporting this hypothesis. Except for minor losses close to the bottom of the well no mud losses were observed during drilling of the open hole section of the well. The location of the well RV-43 is shown in Figure 1.

The first and only stimulation of RV-43 took place in 2001 after its drilling. Water of pressure up to 10 MPa was injected along the open-cased segment of the well, and the total injected volume was not documented. However, this can be inferred from the original drilling report, which states that at least 1,900 m$^3$ were injected and no seismicity observed (suggesting a maximum magnitude threshold $M < 2$). After the stimulation, the well had an injectivity index less than 6·10-9 m$^3$ /Pa·s for the maximum injection's pressure, which is at best half of the required value for commercial exploitation.

### 2.2 State of stress and structural geology

A first estimate of the state of stress at Geldinganes has been inferred from a global Icelandic stress survey conducted by Ziegler et al., 2016 and by Heidbach et al., 2016. They suggest a potential orientation for $\sigma_{Hmax}$ of 340˚-40˚ NW-SE, based on 12 geological indicators in a 10 km region around the site. The magnitude of the stress at depth could be extrapolated from shallow hydrofracturing stress measurements. Such tests were conducted in two boreholes (H32 and H18) near Reykjavik on the flank of the Reykjanes-Langjokull continuation of the Mid-Atlantic Ridge (Haimson and Voigt, 1976; Haimson, 1978).

Four tests were conducted in the borehole H32 between 200 and 375 m depth in jointed basalt. The minimum compressive stress, $\sigma_{Hmin}$, was found to be horizontal in the range of 4 to 6 MPa, while for the maximum horizontal stress, $\sigma_{Hmax}$, it was approximated as varying between 5 to 10 MPa for the four tests. The direction of $\sigma_{Hmax}$ was calculated based on three hydro-fractures with an orientation N25°W ± 5°. The vertical stress, $\sigma_V$, was calculated based on a gradient of 27 MPa/km. These values, if extrapolated to a depth of 1.5 km, suggest a normal stress regime. In the borehole H18 only three tests were performed due to extensive jointing. While the test at 180 m was conducted in basalt, the lower tests at 290 and 324 m were in an intrusive dolerite. Also, in this case, the minimum principal stress was found to be horizontal ($\sigma_{Hmin}$) increasing with depth from 4 to 8 MPa. For the maximum horizontal stress ($\sigma_{Hmax}$) it was observed in a range from 12 to 16 MPa, while the vertical stress





ranged from 5 to 9 MPa. For these tests, the hydro-fractures suggest contradictory directions, and hence two possible orientations

for $\sigma_{Hmax}$: N20°E for the 180m test, and N45°W for the 290m test. Extrapolation of the results at 1.5 km depth suggests in this

case a strike-slip/thrust regime.

        Combining the results of both boreholes, a linear approximation of the data between 200 m and 350 m depth gives

$\sigma_{Hmin}$ = 21 MPa/km, $\sigma_{Hmax}$ = 3 MPa + 30 MPa/km, and $\sigma_V$ = 27 MPa/km. As reported by Haimson and Voigt, 1976, the

measured stress orientation (H32) has no obvious relationship to the NE strike of individual rift zone fissures and faults, inferred

WNW direction of lithospheric plate motion, or axial rift zone earthquake focal solutions which indicate NW-trending. The

measured stresses could be related to (i) a hot spot, (ii) local phenomena involving the extinct NNW-trending Kjalarnes central

volcano, or (iii) ground distortion due to fluid withdrawal from the Laugarness hydrothermal system. Finally, the two stress

measurements in dolerite in borehole H18 could be interpreted as high stress layers. By excluding these two measurements, the

linear approximation gives $\sigma_{Hmin}$ = 2 MPa + 10 MPa/km, $\sigma_{Hmax}$ = 3 MPa + 13 MPa/km, and $\sigma_V$ = 27 MPa/km, leading to normal

conditions at 1.5 km depth (Hofmann et al., 2020).

## 2.3 Planned activity

The re-stimulation of well RV-43 is foreseen by the end of October 2019. The re-stimulation is based on a three-staged cyclic

pulse stimulation that will last for circa 12 days (4 injection days per stage). In particular, it is expected to enhance productivity

in three pre-existing fracture zones penetrated by RV-43 and isolated with straddle packers. Specifically: (i) the first zone is

located at 1700-1750 m Measured Depth (MD) that corresponds to 1467-1507 m in True Vertical Depth (TVD), where the basalt

intersects with the gabbro and mud losses had been observed, (ii) the second zone is located at 1300-1350 m MD (1150-1189 m

TVD), (iii) the third zone is located at depth 1100-1150 m MD (1001-1032 m TVD). Each of the zones will be stimulated with a

cyclic injection scheme ("cyclic" stimulation), which repeats every 24h and includes pressurizing RV-43 with pulses of frequency

1/60 Hz ("pulse" stimulation) and continuous injection phases. This is illustrated in Figure 2.

        The application of short-term cycles is based on the concept of fatigue-hydraulic fracturing, introduced by Zang et al.

2013; 2017; 2018. In practice, pressure pulses are expected to weaken the rock ("fatigue") by inducing microcracks before

macroscopic failure. This mechanism has three major intended benefits: first, the stimulated reservoir volume is increased due to

more complex fracture growth, and a larger and denser fracture network provides a larger heat exchanger area; second, the

breakdown pressure is reduced, therefore, lower injection pressures are required to stimulate the target formation hence reducing

the potential for slip on faults and, thus, the likelihood of induced seismic events; third, the magnitude of the largest induced

seismic events is potentially limited. The concept of cyclic stimulation is one of the soft stimulation techniques evaluated by the

DESTRESS consortium (Pittore et al., 2018). The stimulation of each stage is expected to last maximum 4 days with the following

schedule including pre- and poststimulation operations:

150        • 1/2 day to install the packer,





- 1/2 day to perform injection tests with stepwise flow rate increase,
- 4 days for the main stimulation (with stepwise flow rate increase, and repeating phases of cyclic injection, cyclic-pulse stimulation and continuous injection),
- 1/2 day for performing flowback, where withdrawn water goes to the sea

- 1/2 day for removing the packer and redressing it for the following stage.

Injected water is not expected to exceed rates of 60 l/s or overpressures of 20 MPa at any time during the stimulation due to restrictions by the equipment. No threshold value for injectivity has been reported for stopping the stimulation and stimulations are expected to continue either until the end of the planned injection or until a Traffic Light System (TLS) forces the termination (e.g., Mignan et.al., 2017). The well will be stimulated sequentially from bottom to top.

160         An exemplary main stimulation for each stage is plotted in Figure 2. During the first day, step rate injection tests are performed for estimating the pressure at which fractures open and to observe the seismic response to increasing flow rates. Based on this, the flow rates of the following phases are determined in order to reach sufficient pressures for stimulation of the target interval. The main part of the stimulation consists of cyclic injection (4 cycles of 1 hr high rate injection and 1 hr low rate injection), cyclic pulse injection (4 cycles of 1 hr high rate injection with pressure pulses and 1 hr low rate injection), and 8 hours of

continuous injection. The volume injected in each of the phases is planned to be (circa) equal. The flow rates depend on the fracture opening pressure. This procedure is repeated up to three times before the flow rates are reduced slowly and stepwise at the end of the treatment.

## 2.4 Mitigation strategy

In the presence of fluid-induced seismic risk, it is paramount to efficiently monitor the induced seismicity and define a risk
mitigation strategy. In the Geldinganes area, a dedicated microseismic network has been recently installed. The seismic monitoring infrastructure, completed in August 2019, consists of 13 seismic stations one seismic array of 7 seismic stations and one deep borehole array of 17 geophones (Figure 1). The stations send data in real-time to the Iceland GeoSurvey (ISOR), which streams them both to  ETH-Zurich and GFZ-Potsdam. Real-time seismic data analysis will be performed using the software package Seiscomp3. Induced seismicity monitoring and risk mitigation operations at Geldinganes are
conducted by a team of experienced professionals, including seismologists, field operation managers, reservoir engineers and an internal expert panel (who will support decision during critical situations). The adopted protocol for the Geldinganes TLS is based on a five-steps action plan that governs the fluid injection operations illustrated in Figure 3 and summarized below.

In the case of induced seismic events above a certain threshold, we require a specific action plan. In particular, we first subdivide the region surrounding the industrial site in an internal and external domain.



- **Internal Domain**: It defines the volume surrounding the industrial operations where seismicity will be monitored and analysed with maximum sensitivity.
- **External Domain**: It is a wider volume surrounding the Internal Domain, where the occurrence of seismicity may still be associated with the industrial operations.

For the Geldinganes site, we set these domains as cylinder-shaped volumes with radii from the injection well of 2.5 km and 5.0 km for the internal and external domain, respectively (Figure 1). The range of depth considered for both domains is between 0 and 10 km. These values have been defined by considering other induced monitoring projects and considering the expected uncertainties for the automated locations. The magnitude of completeness of the Geldinganes network, evaluated using the BMC method (Mignan et al., 2011; Panzera et al., 2017), is ~0.3 and ~0.0 for the external and internal domain respectively. Seismic events with $M_L > 0.0$ and occurring within the internal domain should be seen clearly by almost all the stations within this domain. Therefore, all the seismic events above this magnitude threshold will be manually analyzed.

For the external domain, we will manually refine the automated solutions for seismic events with $M_L > 0.5$. Since automatic magnitudes of small events might be overestimated, these thresholds need to be revised by considering the observed seismicity data collected during the early stage of stimulation operations. Then, the analysed data are used to update the risk study and to assess the performance of the monitoring network. These analyses are performed at the early stage of the cyclic stimulation, and injection is increased carefully until at least a few events are detected and located. For seismic events occurring within the internal or external domain, the following alert levels are defined:

- **No Alert**: Seismic events with local magnitude below 1.0 either in the internal or external domain do not trigger an alert. *Seismic analysis:* The routine seismologist manually reviews every day the seismic events that occurred in the past 24h and within the two domains. For the internal domain, all the events with magnitude $0.0<M_L\leqslant1.0$ must be manually reanalyzed, while for the external domain the magnitude range is $0.5<M_L\leqslant1.0$. *Field operations:* Operations continue as planned, if no anomalous seismicity patterns are identified. Otherwise, the internal expert panel is informed and after an update of the risk assessment, specific case-dependent actions are taken (Zoback, 2012). *Communication:* If the seismologist on duty identifies anomalous seismicity patterns in the space-time-magnitude domain, the routine seismologist report to the internal expert panel who decides the actions to take in conjunction with the operation management.
- **Green level**: This level is activated if one or more earthquakes within the magnitude range $1.0<M_L\leqslant1.5$ occur in the internal or external domain. It can be considered as a pre-alert level. *Seismic analysis:* No immediate response is required by the seismologist. However, within few hours (4-6 hours) the routine seismologist reviews the overall characteristics of seismicity (e.g. rate and b-value changes in the Gutenberg-Richter law), including its spatial and temporal evolution during the last 24h. Most importantly, the routine seismologist must check the presence of



seismicity patterns resembling potential faults by using more sophisticate techniques. *Field operations:* Operations continue as planned, if no anomalous seismicity patterns are identified. Otherwise, the internal expert panel is informed and after an update of the risk assessment, specific case-dependent actions are taken (Zoback, 2012). *Communication:* If anomalous seismicity patterns are found, the routine seismologist must inform the internal expert

panel and the field operations manager.

- **Yellow level**: This level is activated if one or more earthquakes within the magnitude range $1.5 < M_L \leqslant 2.0$ occur in the internal or external domain. *Seismic analysis:* Immediate response is required by the seismologist on duty, and within 45 minutes a manual location and magnitude estimation is sent to the operators. Consistency with the Icelandic Meteorological Office (IMO) magnitude needs to be checked. These events, if very shallow (i.e. 1-2 km), can be

potentially felt by people living close to the epicenter. *Field operations:* Flow and pressure decrease until seismicity levels remain below the green alert for at least 4 hours. *Communication:* Status report sent to the IMO.

- **Orange level**: This level is activated if one or more earthquakes within the magnitude range $2.0 < M_L \leqslant 3.0$ occur in the internal or external domain. *Seismic analysis:* Immediate response is required by the seismologist on duty, and within 45 minutes a manual location and magnitude estimation is sent to the operators. Consistency with the IMO

magnitude needs to be checked and, in this case, IMO magnitudes are the preferred estimate that are sent to the operators. These events can be felt, depending on the depth of the event and the distance of the epicenter from urbanized areas. *Field operations:* Stop injection operations and bleed-off of the current stage. Continue with stimulation of other stages if seismicity remains below the green level for at least 12h. *Communication:* Status report sent to regulators, the ICPD, IMO and RE's public affairs.

- **Red level**: This level is activated if one or more earthquakes with local magnitude $M_L > 3.0$ occurs. *Seismic analysis:* Immediate response is required by the seismologist on duty, and within 45 minutes a preliminary manual solution is produced. For magnitude estimation, since the closest ones to the epicenter might be saturated, stations with a distance of 10 km from the epicenter are used for the calculation. IMO magnitudes are the preferred and most reliable estimate in this case. These are felt and potentially damaging events. *Field operations:* Immediately stop injections operations

and bleed-off taken. The stimulation operations for this well will be discontinued and not started again. *Communication:* Status report sent to regulators, the ICPD, IMO and RE's public affairs, asses potential damage.

## 3 Probabilistic fluid-induced seismic hazard assessment

Probabilistic risk assessment is emerging as the standard approach to manage and mitigate induced seismicity linked to fluid injections in the underground (Mignan et al., 2015; 2017; 2019b; Bommer et al., 2017; Broccardo et al., 2017a; Grigoli et al.,

2017; Lee et al., 2019). The need for a probabilistic risk-based approach is motivated by the stochastic nature of earthquakes, the



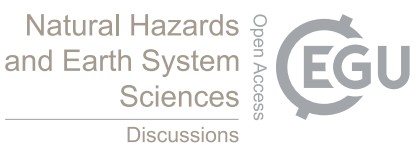

many uncertainties associated with the process of inducing seismicity and the needs of regulators, insurance and the public (Mignan et al., 2019a; 2019b). Both hazard and risk approaches follow standards proposed, among others, by the Swiss Seismological Service (SED, 2017) and related references (Broccardo et al. 2017a, Mignan et al., 2015; 2017; 2019a; b), which are based on a combination of Probabilistic Seismic Hazard Analysis (PSHA) and the PEER-PBEE framework (Cornell, 1968;

Cornell, Krawinkler, 2000).

PSHA is assessed as the probability of exceeding a given intensity at a given distance $R$ from the injection site, based on the number of events above a given minimum magnitude $m_0$, the frequency distribution of the magnitude (namely the truncated Gutenberg-Richter distribution), and an empirical ground shaking attenuation function. The latter can be an intensity prediction

equation (IPE) based on felt intensity, or a ground motion prediction equation (GMPE) based on peak ground acceleration (PGA), spectral acceleration (SA) or peak ground velocity (PGV). Commonly, within this probabilistic framework, there are two main elements to be defined: (i) the probabilistic characterization of the seismogenic source model(s), and (ii) the ground motion characteristic model(s) (describing the expected ground vibration given the occurrence of an earthquake). The first gives the temporal and spatial forecast of the earthquake ruptures, while the second is characterized by GMPEs to link the earthquake

rupture with the expected ground shaking at the site of interest.

The output of PSHA analysis is the rate of exceedance or hazard curves (probability of exceedance for a given period of time) of a given ground shaking Intensity Measure ($IM$) type. A single curve (for a given set of parameters) represents the aleatory (irreducible) variability within the defined model. To include also the epistemic uncertainties, given the alternative possible models, a logic tree structure with weighted branches (indicating the belief in a given model) is defined (e.g., Mignan et al., 2015).

Figure 4 shows the proposed logic tree adopted for this a-priori risk analysis. The first level of the logic tree defines the seismogenic source models, the second level the upper bound of the Gutenberg-Richter distribution, the third level the GMPEs and the ground motion intensity conversion equations (GMICEs). In the following, we report a detailed discussion for each level of the logic tree.

## 3.1 Seismogenic source models

In this analysis, we assume that induced seismicity nucleates and eventually extends in the proximity of the injection point. Therefore, a point source located at the coordinates of the injection point is used as the unique seismogenic source model for the investigation. This implicitly excludes any geometrical uncertainty on the location of the hypocenter. Forecasting the number of events that will occur in a reservoir stimulation is difficult because (as previously stated) the stressing conditions and location of faults near the injection point are unknown. Empirical data from similar sites can be used as a first-order proxy, but in the

case of Geldinganes, only limited experience exists. In light of these limitations, we argue that the spatial variability of the seismicity is well constrained by a simple seismogenic source that can be updated for real-time application.





The number and size of earthquakes in PSHA analysis is based on three parameters that describe the local seismic activity rate, the event size distribution, and the largest event size (Cornell, 1968). These parameters are typically constrained based on observed seismicity with the activity rate broadly scaling with the seismo-tectonic strain input. For induced seismicity, the seismogenic models likewise must also describe the local seismic productivity that is (in this case) linked to the injection profile. Such seismicity rates are unknown, albeit the hydraulic energy input might be estimated beforehand. Moreover, the link between induced seismicity and stress release is a key factor to be considered in the analysis. The fraction of seismic to hydraulic energy may thus vary from zero (no events observed) to well above 1 (sometimes referred to as "triggered" events that release mostly pre-accrued tectonic stresses, e.g., Pohang). We will consider two simple seismogenic source models to analyze this uncertainty in energy release and have a first-order forecast of the underground response to injection:

- **Model SM1:** A seismogenic source model that assumes the underground feedback is site-specific constant, with all parameters purely data-driven (Mignan et al., 2017; Broccardo et al., 2017a).

- **Model SM2:** A seismogenic source model that simulates the fluid and overpressure propagation for the planned injection protocol based on one-dimensional diffusion and stochastically distributed seeds. Model SM2 will also explicitly use the observation of no seismicity at M >= 2 during the first stimulation in the year 2001 as a constraint. The synthetic catalogues are then converted onto the same underground feedback site-specific parameters of Model SM1 (Karvounis et al., 2014; Karvounis and Jenny, 2016).

These two models capture to a first-order the epistemic uncertainty in forecasting seismicity, since they express alternative approaches to forecasting (purely empirical and partially physics-based). Both models are equally weighted to estimate the ground shaking estimation at the site of interest.

### 3.1.1 Seismogenic source model SM1

The seismogenic source model SM1 assumes that the "seismic underground feedback" per volume affected by significant pore-pressure change is a site-specific (and generally a-priori unknown) constant. This constant can vary by several orders of magnitude between sites. Because the volume affected scales with the volume of fluid injected (and in theory to the pressure applied; Mignan, 2016), this implies a relation between the expected number of earthquakes $E[N]$ and the volume injected $V$, as

$$E[N(t); M > m] = \begin{cases} 10^{a_{fb}-bm}V(t), t \le T_{in} \\ 10^{a_{fb}-bm}\tau \exp\left(-\frac{t-T_{in}}{\tau}\right)\dot{V}(T_{in}), t > T_{in} \end{cases}, \tag{1}$$

where $a_{fb}$ is the underground feedback parameter (i.e., the overall activity for a given volume $V$, which is also known as seismogenic index $\Sigma$; Dinske and Shapiro, 2013), $b$ is the slope of the Gutenberg-Richter distribution, $T_{in}$ is the injection duration,





and $\tau$ is the mean relaxation time of a diffusive process. This relation (during the injection phase) is broadly accepted in the

technical community as a first order model (e.g., Dinske and Shapiro, 2013; van der Elst et al., 2016; Mignan, 2016; Broccardo

et al., 2017a). The post injection phase has been added by Mignan et al., 2017, to account for the decrease of the rate of seismicity

after the injection has been terminated (trailing effect) (Mignan, 2015). This model has been verified for a number of fluid injection

experiments, in terms of flow rate $\dot{V}$ versus induced seismicity rate $\lambda(t, M > m)$ (Mignan et al., 2017). This allows a refined

analysis by defining the rate function

$$\lambda(t, M > m) = \begin{cases} 10^{a_{fb}-bm}\dot{V}(t), \text{for } t \leq T_{in} \\ 10^{a_{fb}-bm}\dot{V}(t)\exp\left(-\dfrac{t}{\tau}\right), \text{for } t > T_{in} \end{cases} \tag{2}$$

which allows to define a Non-Homogeneous Poisson Process (NHPP). Note that this model only applies to the stimulation phase

in which the fluids injected are not supposed to be produced back, hence creating an overpressure field at depth z. In practice,

after each stimulation stage parts of the injected fluid will be produced back by natural bleed-off (without pumping) directly after each

stage and by airlift testing after the end of the last stage. Note also that $E[N(t), M > m] = \int_0^t \lambda(t, M > m)dt$ and the expected total

number of fluid-induced earthquakes is $E[N(\infty); M > m] = 10^{a_{fb}-bm}\left(V(T_{in}) + \tau\dot{V}(T_{in})\right)$.

While the parameters $a_{fb}$, $b$ and $\tau$ can be estimated during the stimulation (Mignan et al., 2017; Broccardo et al.,

2017a), a-priori knowledge on those parameters is limited and the range of possible values wide. We list the $a_{fb}$, $b$ parameter

estimates for different sites in Table 1, which will be used as input for the a-priori risk study. Uncertainties are likely to

significantly reduce once seismic data is obtained by monitoring during the stimulation. Note that due to the correlation of $a_{fb}$

and $b$ (Broccardo et al., 2017a), pairs of ($a_{fb}$, $b$) values from different sites need to be maintained. In Mignan et al. (2017), the

mean relaxation time has been observed to widely vary between injection sites with $0.2 < \tau < 15$ days.

In order to apply a classical PSHA analysis, we transform the NHPP into a Homogeneous Poisson Process (HPP), using the

equivalent rate $\Lambda_{M>2} = E\left[N(T_{in} + T_{p_{in}}), M > 2\right] = \int_0^{T_{in}+T_{p_{in}}} \lambda(t)dt$, for a unit of time which corresponds to the total project

period (including the post injection phase), i.e., $T_{in} + T_{p_{in}}$, where $T_{p_{in}}$ is the post injection time. We selected $M > 2$ because we

assume that lower magnitudes will not have the potential to trigger any damage. By doing so the $P\left(M > 2; T_{in} + T_{p_{in}}\right) = 1 -$

$\exp\left(-\Lambda_{M>2}\right)$. Table 1 and Figure 5 (red dots) report the equivalent rate $\Lambda_{M>2}$ for each project, for a target injected volume of

circa V = 18,000 m3 (estimated from c. 6,000 m3 injection per stimulation, times 3 stimulations; Figure 2). At the present time,

without any pre-stimulation phase it is not possible to infer where the Geldinganes project is placed in this domain; however, what

is known is that 5000 m³ of water were injected and no seismicity observed.

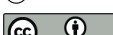



### 3.1.2 Seismogenic source model SM2

With model SM2, a first-order physical process is included into the forecasting. This is done by modeling pressure diffusion through a fractured media containing randomly distributed earthquake faults (so called "seeds"). The pressure propagation can
be adopted based on the reservoir properties, limited to the available information. Then, the density of these seeds and their size distribution are treated as free site-specific parameters that (again) are unknown a-priori. These models are commonly referred to as "hybrid" models (Gischig and Wiemer, 2013; Goertz-Allman and Wiemer, 2013) as they combine deterministic and stochastic modelling. Specifically, the adaptive Hierarchical Fracture Representation (a-HFR) is employed both for modeling flow in a fracture network with dynamically changing permeability (Karvounis and Jenny, 2016) and for simulating
the source times of randomly pre-sampled scenarios of hydro-shearing events at certain hypocenters (Karvounis et al., 2014). This hybrid model is chosen here, as it can integrate several of the field observations, returns forecasts both of the spatial distribution of seismicity and of its focal planes, and can forecast reservoir properties like the expected well's injectivity at the end of the injection.

The required inputs for the proposed hybrid model are the initial hydraulic properties, the planned activities, a first-
order knowledge of the stress conditions in the proximity of the well, and the orientations of pre-existing fractures. Here, the extrapolated stress measurements described in Section 2.2 are employed without excluding any of the measured stresses; i.e., the vertical direction is a principal direction and $\sigma_{Hmin}$ = 21 MPa/km, $\sigma_{Hmax}$ = 3 MPa + 30 MPa/km, and $\sigma_V$ = 27 MPa/km. The planned activities are described in Section 2.3, the initial transmissibility is in agreement with the currently expected injectivity of $6 \cdot 10_{-9} \, m^3/(Pa \cdot s)$ along the whole open segment of RV-43, and the compressibility is inferred from the nearby
well HS-44, since there are no reported measurements of the characteristic time at RV-43. Moreover, in this a-priori analysis, all surface orientations are considered equally probable. Observe, however, that two distinct sub-vertical fault sets seem to prevail at the mainland surrounding the bay above the injection stages (Hofman et al., 2020).

We assume a constant value for those parameters for which the rate of seismicity is less sensitive. These are: the friction and the cohesion of fractures for the Mohr-Coulomb failure criterion which are fixed equal to 0.6 and 0, respectively,
and the mechanical aperture of fractures after they have slipped is 1 mm. The remaining parameters (those for which seismic rate is more sensitive) are the spacing between pre-existing fractures, the $b$-value of the Gutenberg Richter during injection, and the permeability of a slipped surface. A range of possible values is considered for each of the latter parameters; i.e., the spacing between pre-existing fractures (1 m, 2 m, 4 m, 8 m, 16 m), the $b$-value of the Gutenberg-Richter law (0.75, 1., 1.25, 1.5, 1.75, 2.), and the hydraulic aperture between the two parallel fracture surfaces (200 μm, 500 μm), out of which aperture
the equivalent permeability can be estimated. A synthetic catalogue of rate $\Lambda_{M>2}$ is created for each possible combination of the above values. The resulting scenarios are shown in Figure 5 as blue dots in $a_{fb}$-$b$ space. Note that all scenarios are above the dashed line that indicates the limit posed by the no-seismicity observation during the initial stimulation.





**3.2 Upper bound for the Gutenberg-Richter distribution vs. Maximum Magnitude distribution**

The frequency magnitude distribution of natural and induced earthquakes follows (to a first order) the classical Gutenberg-
Richter distribution. This distribution is truncated at an upper-end for energy conservation, but also because existing faults and
fault systems have a maximum size (e.g., Mignan et al., 2015). Empirical observations will only poorly constrain the largest
possible earthquake, since it is by definition an exceptionally rare and extreme event.  One of the major sources of uncertainty
is thus in PSHA related to the upper bound of the (truncated) Gutenberg-Richter distribution, here indicated as $m_{sup}$. However,
it is also true that in most seismic hazard and risk studies, the actual value of $m_{sup}$ is not a critical choice, since its rate of
occurrence is typically very small compared to the typical return periods of interest. It follows that (in general) both hazard
and risk are typically dominated by the more frequent, moderate-size events.

It is generally accepted that the largest possible induced earthquake cannot be larger than the tectonically largest one.
However, in induced seismicity, the tectonic environment (controlled primarily by the state of stress) at a site may be such that
no tectonically prestressed larger ruptures exist. Under these conditions, ruptures will be running out of energy once they leave
the volume brought into a critical state for failure by the injection—e.g., because of the effect of pore pressure on the Coulomb
failure criteria. In these conditions, run-away ruptures cannot occur even if a natural fault exists (also referred to as "triggered"
events cannot happen), and the largest magnitude size as a consequence is limited by the volume or area affected by
overpressure, which again scales with the volume of fluid injected (or extracted) and the hydraulic properties of the subsurface.
In such situation, $m_{sup}$ can locally be substantially smaller than the regional tectonic one. This is common in "fracking"
operations in tight shales. McGarr (1976; 2014) formalized this volume limit as $m_{sup,McGarr} = 2/3 \log_{10}(GV) - 10.7 + 14/3$ where
$G = 3 \cdot 10_{10}$ Pa is the modulus of rigidity. McGarr has shown that this relationship is consistent with the data from a compilation
of injections. However, a number of researchers (Gischig et al., 2014; van der Elst et al., 2016; Mignan et al., 2019b) have
pointed out that outliers exist (e.g., Pohang, South Korea, Grigoli et al., 2018, Kim et al., 2018; St. Gallen, Switzerland, Diehl
et al., 2017) and that the McGarr limit is best explained as a purely statistical relationship based on simple extreme value theory
principles (Embrechts, 1999).

The "McGarr limit" has been used (and in some cases one might argue misused) in numerous induced seismicity
hazard assessments (van der Elst et al., 2016). For $V = 18,000$ m3, we would for example obtain $m_{sup,McGarr} = 3.79$. Based
on the recent statistical tests of van der Elst et al. (2016) and the occurrence of the 2017 Pohang  earthquake above the expected
limit (Grigoli et al., 2018, Kim et al., 2018), a fixed McGarr limit now appears questionable to many seismologists. Therefore,
since the complete information about the number, location, size and stressing condition of faults in the Geldinganes area is not
available (in particular before the stimulation phase), it is appropriate to consider $m_{sup}$ as the regional tectonic $m_{sup} = 7$
(Kowsari, et al. 2019). This estimate could be reduced at a later stage if local fault information were found to provide better
constraints.





In addition, in this study, we determine the probability distribution of the maximum magnitude, $M_{max}$, observed at
fluid injection sites for the total time of observation. This is fundamentally different from the upper bound $m_{sup}$ of the
Gutenberg-Richter distribution, which is merely a deterministic upper limit fixed by physical constrains. The probability
distribution of the maximum magnitude, $M_{max} = max\left[M_1, ..., M_{t_i}, ..., M_{T_{in}+T_{p_{in}}}\right]$, can be easily derived considering the
magnitude events statistically independent. It follows that $F_{M_{max}}(m|N = n) = F_M(m)^n$, where $F_M(m)$ is the classical
Gutenberg-Richter cumulative probability density function, and $f_{M_{max}}(m|N = n) = nF_M(m)^{n-1}f_M(m)$. Since the number of
events is a random variable itself, then $F_{M_{max}}(m) = \sum_n F_{M_{max}}(m|N = n)P(N = n|\Lambda(T_{in} + T_{p_{in}}))$, $f_{M_{max}}(m|N = n) =$
$\sum_n nF_M(m)^{n-1}f_M(m)P(N = n|\Lambda(T_{in} + T_{p_{in}}))$, where $P(N = n|\Lambda(T_{in} + T_{p_{in}}))$ is the classical Poisson discrete distribution.
Figure 6-a,b shows the equivalent rate of seismicity, $\Lambda(T_{in} + T_{p_{in}})(1 - F_{M_{max}}(m))$, (i.e., a weighted  complementary CDF)
for each of the project reported on Table 1 (SM1 model) and for each of the synthetic catalogues (model SM2). We can observe
a large scatter of rate of seismicity reflecting the large uncertainty exiting prior to the project.

Together with the distribution of $M_{max}$ for each $a_{fb} - b$ couple, we report the envelope distribution computed as the
mean value over all the branches of the logic tree (Figure 4). Figure 6-c shows the envelope distribution. Observe that given
the sparse dataset (Table 1), this distribution is (inevitably) multimodal. The expected $E[M_{max}]$, based on this envelope
distribution, is 2.25 and the 5-95% interval is $[0.10 - 4.45]$. It is important to highlight that these values represent some
statistics based on previous projects and not the expected values for this project. In fact, here, the envelope distribution
represents a prior distribution, which must be updated during a pre-stimulation phase and during the stimulation. In the
following, we also report the envelope distribution of $M_{max}$ based on the synthetic catalogue derived according to the SM2
source model. Figure 6-d shows the envelope distribution. Different from the envelope distribution based on the SM1 source
model, this distribution shows a more regular shape, since the synthetic dataset is denser and more confined. However, this
prior distribution can be affected by overfitting since it is based on stress measurements (without considering their
uncertainties) that might not represent the current local condition correctly. The expected $E[M_{max}]$, based on this envelope
distribution, is 2.09 and the 5-95% interval is $[1.19 - 3.42]$. Finally, Figure 6-e we reported the $E[M_{max}]$ and $[5-95]\%$
confidence bound as function of the injected volume.

### 3.3 Ground Motion Prediction Equations and Intensity measures

The relationship between the site source characteristics and given ground shaking intensity measure types, $IMs$, is
given by seven ground motion prediction equations (GMPEs). Kowsari et al. (2019) provide a set of adjusted GMPEs that
have been selected for this investigation. In particular, the proposed GMPEs were adjusted using newly compiled ground
motion records of six strike-slip events in the South Iceland Seismic Zone (SISZ), with a range of magnitudes between $M \in$





$[5, 6.5]$ ($M$ is intended as $M_w$ as used in Kowsari et al. 2019), and distance $R \in [0,80]$ km. The intensity measures are reported in Table 2, and the value of the functional form and the coefficients can be retrieved directly from Kowsari et al. (2019).

Observe that from the original list we replaced the proposed GMPE of Lin and Lee (2008) for North Taiwan with the local GMPE (RS09), Rupakhety and Sigjörnsson (2009), which is consistent with the strike-slip nature of Icelandic earthquakes. The recalibration has been performed only for the PGA; therefore, in the following, we assume only this physical intensity measure. The selected site-to-source distance is the Joyner-Boore metric ($R_{JB}$) (i.e., the closest horizontal distance to the vertical surface projection of the fault). When the distance metric of the original GMPE is different from $R_{JB}$, the same

transformations proposed in Kowsari et al. (2019) are applied. In the Supplement, Figure S1, we show the trellis plots for the selected GMPEs models.

It is important to highlight the limitation of these choices. First of all, the GMPEs are calibrated for natural events that are considerably larger in magnitude compared to the expected fluid induced events (Figure 6). Therefore, the extrapolation to lower magnitudes is probably biased. This will have a more significant effect on the low damage threshold, while the $IR$

computations are less impacted since they depend on larger events. Moreover, for small events in the proximity of the injection point, the ideal source-to-site distance is the hypocentral distance and not $R_{JB}$ (observe that in this case $R_{JB}$ converges to the epicentral distance), which neglects the hypocenter depth. As a consequence, this analysis is independent of the injection depth. Again, this limitation has an impact on the small damage threshold, since the depth of the events is expected to have a significant influence.

In this a-priori assessment, we use as final $IM$ the European Macroseismic Scale (EMS98, Grünthal, 1998). The advantage of EMS98 over physics-based intensity measures, in this phase, lies in the easier interpretability of this scale, which is based merely on shaking indicators expressed in terms of damage and nuisance to the population. Based on these considerations, the selected GMPEs are converted into expected intensity by using GMICEs for small-medium intensities. The GMICEs used in this work are introduced by Faccioli and Cauzzi (2006) and Faenza and Michelini (2010). The aleatory variability is then

combined into a GMPE-GMICE model with $\sigma_{TOT}$ defined as $\sigma_{TOT} = \sqrt{(\sigma_{GMPE}^2)a^2 + \sigma_{GMICE}^2}$, and values of the mean, $\sigma_{GMPE}$, $\sigma_{GMICE}$, and $a$ reported in the Supplement together with the combined trellis plots (Table S1, Figure S2).

### 3.4 Probabilistic hazard results (PGA, EMS98)

The hazard integral is reduced to the marginalization of the random variable magnitude, $M$, and the conditional random variable $IM|M = m$, since the site-to-source distance is fixed by the source point (which is assumed at the injection point). For a given

site, then the rate of exceedance is simply reduced to $\Lambda(im; T, b) = - \int_m P(IM > im | M = m, r) \, d\Lambda_{M>2}(m; T, b)$, where $d\Lambda_{M>2}(m; T, b) = \Lambda_{M>2}(T) \, F(m)$, with $F(m)$ equal to the Gutenberg-Richter above magnitude 2. Given the discussion in Section 3.1.1, the probability of exceedance of an intensity, $IM = im$, for a given time period (which corresponds to the total

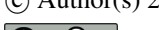



duration of the project given the normalization introduced in Section 3.1.1) is given by the Poisson distribution as

$$P(IM > im, t = T) = 1 - \exp(-\Lambda(im; T, b)).$$

$\Lambda_{M>2}(T)$ is not known a-priori (neither an uncertainty quantification based on local condition can be carried out a-priori), therefore we compute the risk for each of the $a_{fb}$ and $b$ pairs of Table 1. As mentioned in Section 3.1.1 the scatter is very large and this reveals the state of deep uncertainty existing prior to a pre-stimulation phase. The PSHA outputs are shown in red in Figure 7 for both the $PGA$, (top panel) and for the $IM$ (bottom panel). These curves confirm the state of deep uncertainty in particular for the location in proximity of the injection point. In fact, for a given probability of exceedance of $10^{-4}$ and distance

2-5 km from the injection point, the macroseismic intensity range between the 10% and 90% percentile is circa $IM \in [6, 11]$. In addition, we report the PSHA analysis based on the source model SM2. The outputs are shown in blue in the same Figures. The epistemic uncertainties of the source model SM1 are considerably higher than the ones arising from the source model SM2. This was expected given the inherent sparsity present in the data of Table 1. Moreover, the epistemic median of the source model SM1 is higher than source model SM2. In addition to the PSHA output, in the Supplement (Figure S3), we reported the hazard-based

scenarios for different magnitude.

## 4 Probabilistic fluid-induced seismic risk

In seismic risk assessment, it is common to distinguish between physical and non-physical risk. Examples (and precedents) of non-physical risk include noise, vibrations felt, opposition by residents, public campaigns against the project, *etc*. Non-physical risk is complex and often impossible to quantify. Therefore, an effective and practical approach should focus on non-physical

risk identification and mitigation rather than risk assessment (Bommer et al., 2015). Conversely, the physical risk faced by exposed communities needs a quantitative assessment. In this study, we focus only on one type of physical risk which is the seismic risk.

      The physical risk is commonly divided into two major categories, i.e., fatalities and/or injuries, and economic losses. The a-priori risk analysis for the Geldinganes project here focuses on the first risk, while the aggregate economic losses are

not directly computed. Here, as a substitute for aggregate losses, we define a low damage threshold for statistical average classes of Icelandic buildings. In particular, in this study, we select two risk measures: Individual Risk ($IR$) and Damage Risk ($DR$). $IR$ is defined as the frequency over the time span of the project (including the post-injection phase) at which a statistically average individual is expected to experience death or a given level of injury from the realization of a given hazard (Jones, 1992; Broccardo et al., 2017b). We here define $DR$ as the frequency over the time span of the project (including the

post-injection phase) at which a statistically average building class is expected to experience light non-structural damage from the realization of a given hazard.





Since there are currently no universally used regulatory and industry approaches to manage induced seismicity of geothermal and other energy projects, we define the following safety thresholds for $IR$ and $DR$. The proposed $IR$ safety threshold is $IR^{ST} = 10^{-6}$. This value is in line with the typical standards for anthropogenic activities for example in

Switzerland or the Netherlands (although in the original definition the time span is a year), and it has been used, for example, in the induced seismicity case of Groningen, Netherlands, (with yearly frequency, van Elk, 2017). In the presence of epistemic uncertainties, the median of the $IR$ distribution is taken as the reference metric to be compared with the selected safety standard, i.e. $q_{IR,.5} \leq IR^{ST}$ (where $q_{IR,.5}$ is the epistemic median of the individual risk distribution). The proposed $DR$ threshold is $DR^{ST} = 10^{-2}$. As for the $IR$, in the presence of epistemic uncertainties, the median of the $DR$ distribution is taken as reference

metric to be compared with the selected safety standard, i.e $q_{DR,.5} \leq DR^{ST}$ (where $q_{DR,.5}$ is the epistemic median of the individual risk distribution).

The framework used for the computation of $IR$ and $DR$ is based on the convolution of vulnerability models for the relevant building typologies with the exposure model. For the fragility-vulnerability model, we should base our analysis on local functions. However, at the present time there exist only local fragility functions for low damage (Bessason and Bjarnason,

2015). Given that, we decide to use the macroseismic intensity approach for $IR$ (Lagomarsino and Giovinazzi, 2006), while using the local fragility function for $DR$.

### 4.1 Individual risk computation

For $IR$, we use a vulnerability given in terms of macroseismic intensity, which follows the macroseismic approach for damage assessment (Lagomarsino and Giovinazzi, 2006) and modified in Mignan et al., 2015, for the induced seismicity case. In this

approach, the vulnerability is not defined based on detailed mechanical models; therefore, it is implicitly assumed that macroseismic and mechanical approaches produce compatible levels of damage. The macroseismic model defines the mean damage grade, $\mu_D(im)$, as function of a vulnerability index $V$, a ductility index, $Q$, and a reduction factor $\alpha$ introduced in Mignan et al., 2015, to recalibrate low damage states to the damage observed in the Basel 2006 sequence. The vulnerability index depends on the building class and construction specifics, and it includes probable ranges $V^-V^+$, as well as less probable ranges $V^{--}V^{++}$.

Following the Icelandic exposure information described in Bessason and Bjarnason, 2016, we select three building typologies: Concrete, Wood, and Masonry as a surrogate for Pumice buildings. Moreover, Bessason and Bjarnason, 2016, observed that (in average) the Icelandic buildings are stronger and more reliable than the ones based in the Mediterranean region in Europe. Based on these considerations, we select $V_0$ as vulnerability index for Concrete and Wood, and $V^-$ for masonry. The choice of $V^-$ for masonry is given by the observation that the fragility of this building is close to old (before the 1980s) Icelandic reinforced

concrete building. Moreover, there is no detailed information on the ductility index for the different class of building, therefore we use $Q = 2.3$, which is the value for masonry structures and reinforced concrete structure with no seismic details. In this phase,



this a practical and conservative choice since $Q = 2.3$ is a lower bound of the possible ranges of values for the ductility index. In the Supplement, we report in Table S2 the vulnerability indices together with vulnerability functions (Figure S4) obtained by using the macroseismic model with parameters $V_0 = 0.5$, and $Q = 2.3$.

We computed the marginal $IR$ considering all $[a_{fb}, b]$ couples in a given location (i.e., different distances), for the total duration of the project (including the post injection phase), for both rate models SM1 and SM2, and using the HAZUS consequence model (Galanis et al., 2018; Hazus MH MR3, 2003). The results are shown in Figure 8 for each building class. Median and quantiles are computed considering a 50% weight for the SM1 model and 50% weight for SM2 model. Despite the median for each class being below the fixed threshold, i.e. $q_{IR,.5} \leq IR^{ST} = 10^{-6}$, the uncertainty is very large; indicating that

uncertainty quantification updates are necessary to reduce the $[a_{fb}, b]$ uncertainties. Moreover, we would like to highlight that the median based merely on SM2 is considerably lower than the median based on SM1 (this is not reported in Figure 8 for clarity). This is due to the lower variability on $[a_{fb}, b]$ arising from the synthetic catalogue.

For $DR$, we use the local fragility model developed by Bessason and Bjarnason, 2016. Three major categories of buildings characterize the Icelandic exposure model: reinforced concrete, timber, and hollow pumice block. Further details on the

exposure model are given in the Supplement (Section S3).  Within these categories, Bessason and Bjarnason, 2016, define the following subcategories:

- Low-rise reinforce concrete
    o **RC-b80**: Reinforced concrete structure designed before seismic code regulations (before 1980).
    o **RC-a80**: Reinforced concrete structure designed after seismic code regulations (after 1980).
- Low-rise timber structures
    o **T-b80**: Timber structure designed before seismic code regulations
    o **T-a80:** Timber structure designed after seismic code regulations
- Hollow pumice blocks (HP)

Fragility functions are provided for all these categories only for small damages (which makes the use for $IR$ impossible). Fragility

functions details and damage-based scenarios for different magnitudes are reported in the Supplement (Section S3).

Finally, we computed the marginal $DR$ considering all the $[a_{fb}, b]$ couples (for both the source model SM1, with weight 50%, and SM2 with weight 50%) in a given location (i.e., different distances), for the total duration of the project (including the post injection phase). The results are shown in Figure 9 for each class of buildings. Again, despite the median for each class being below the fixed threshold, i.e. $q_{DR,.5} \leq DR^{ST} = 10^{-2}$, there is a need for uncertainty quantification updates to reduce the

$[a_{fb}, b]$ uncertainties.



**4.2 Sensitivity analysis**

In this Section, we describe a sensitivity analysis of the epistemic uncertainties with respect to the Quantities of Interest (QoI) $IR$ and $DR$. Specifically, we analyze the sensitivity to the earthquake rate model and the GMPE (and GMICE), which (here) are the only source of epistemic uncertainty. The goal of this sensitivity analysis is to calculate which source of the two input uncertainties is dominant. Specifically, we performed two sensitivity analysis, one for the dataset in Table 1, and one for the synthetic catalog. This allows to better understand the relative contribution of the input uncertainties for each dataset.

In this study, we adopt a screening method which aims to a preliminary and qualitative analysis of the most important input parameter. In particular, we develop a modified version of the Morris method (Morris, 1991), which solves some drawbacks of the "tornado diagram" (Porter et al., 2002) used in Mignan et al., 2015. A "tornado diagram" is a type of sensitivity analysis based on a graphical representation of the independent contribution of each input variable to the variability of the selected QoI. Specifically, given a base model, for each considered variable, we estimate the maximum positive and negative swing of the QoI while holding all the other parameter fixed to their base value. A drawback of the method is that results are strongly dependent on the base model (i.e., it is a local sensitivity method). Therefore, we introduce a variation of the method to obtain a global sensitivity measure. The complete details of the introduced method are reported in the Supplement (Section S4), while here we discuss the general principles and the results. To obtain a global sensitivity measure, we first define a normalized local sensitivity measure of the parameter $i$ with respect to the base model $j$, $d_i(j)$ (Eq. S1 of the Supplement). Then, we define two global sensitivity measures: the average, $\mu_{d_i}$, and the maximum, $\bar{d}_i$, of $d_i(j)$ (Eq. S1 and S2 of the Supplement). The sensitivity measure $\mu_i$ describes the average relative contribution of the parameter $i$ over all possible base models $j$. The sensitivity measure $\bar{d}_i$ describes the maximum contribution of the parameter $i$ over all possible base models $j$. The two measure in this form are not normalized to one.

Given $IR$ and $DR$, Figure 10 and Figure 11 show the sensitivity results based on $\mu_{d_i}$ for each building class. Both measures show the same pattern. In particular, the dominating source of uncertainty is (as expected) the rate model. It is interesting to remark that the rate model contribution is more dominant for the dataset based on Table 1 than for the synthetic dataset. Moreover, this fact is consistent across different building typologies an risk metrics. This corroborates the observations that we have previously made, i.e. the uncertainties related to real data are larger than the synthetic ones (which might be affected by overfitting).

The same trend is observed for $\bar{d}_i$ (Figure S8 S9 in the supplementary material).

**5 Discussion and Conclusion**

In this study, we have attempted to combine all available risk-related information on the upcoming stimulation (middle October 2019) of the RV-43 well in Geldinganes into one quantitative assessment. Our key objectives were:



- **Interdisciplinarity-based risk:** We integrated hydraulic reservoir modelling, empirical data of past sequences, expert knowledge, ground motion prediction equations, as well as exposure and vulnerability information into one quantitative risk assessment. Thereby, in this study, we tried to represent the "center, body and range of the informed technical community," a key requirement for seismic hazard and risk assessment (NUREG/CR-6372, https://www.nrc.gov/reading-rm/doc-

collections/nuregs/contract/cr6372/vol1/index.html).

- **State of knowledge:** The methodologies applied represent the current state of knowledge in earth-science and engineering. They are beyond the commonly adopted state of practice in geothermal projects, but well aligned with good practice recommendation of the DESTRESS project (Grigoli et al., 2017), with Swiss good practice recommendations (Trutnevyete and Wiemer, 2017) and the recommendations of the international expert committee investigating the Pohang earthquake (Lee

et al., 2019).

- **Explicit uncertainty treatment:** We systemically considered the uncertainties in knowledge and the variability of the data in our assessment and made them explicit through the use of a logic tree approach. This reflects the current state of practice in probabilistic seismic hazard and risk assessment for natural earthquakes.

- **Transparency and Reproducibility:** This study documents all decisions taken in a transparent and reproducible way. All

stakeholders in risk governance thus have access to the same level of information as baseline, and ideally a common understanding of the project's risks.

- "**Updatable:**" Most important, the a-priori risk assessment can be updated in a consistent way as soon as new data arrives. Because the initial uncertainties are very large (e.g., Figure 8), updating it with in-situ information is a must and should be done in a manner which is fully compatible with the initial risk assessment. The a-priori risk assessment presented here is thus

also a first and critical step toward risk management.

### 5.1 Limitations of our study

Probabilistic risk assessment is in many ways a very pragmatic approach that systematically collects available information based on the current state of knowledge. It is acceptable that in many areas, the state of knowledge is limited and evolving. While we consider the current assessment as useful and usable, there are also some limitations and areas where further improvements would

be beneficial:

- Geological and seismotectonic knowledge is poorly represented. This is mostly a consequence of the fact that knowledge of the local seismotectonic is limited and uncertain, especially when extrapolated to the reservoir depths. The limited use of





geological constraints is also a consequence of the fact that geological knowledge cannot be readily transferred into forecasting models of seismicity.

• Empirical data from similar injections in the surroundings of Geldinganes or from areas with comparable conditions are limited and mostly based on observation in the 1970's with limited seismic monitoring in place. While countless well-monitored injections have been conducted in Iceland overall, there has been less activity near Reykjavik. The initial stimulation of the Geldinganes well in 2001 produced no noticeable seismicity, which provides important constraints (Figure 5). However, monitoring then was at that time quite limited so smaller than magnitude 2 event may have been undetected and we need to 600 consider also that the response to the 2019 stimulation may also be different.

• The seismicity forecasting models we use are simplistic in many ways, considering a limited amount of physical, hydraulic or geological aspects. In particular, neither the SM1 nor the SM2 model explicitly consider the (re)activation of the cracks/faults responsible for the mud losses during drilling and reported by Steingrimsson et al., 2001. We also use few models overall and do not take the risk-limiting effect of mitigation measures explicitly into account.

• Ground motion models which are specific for Iceland exist. However, they originate from few strong-motion data at short distances, from larger magnitudes and from natural earthquakes and are therefore a limited constraint. Likewise, little is known about the site amplification at a microzonation level. We do not plan in the Advanced Traffic Light System (ATLS) to update ground motion models.

• Building vulnerabilities are known at a first order level, but no efforts have been made to verify or validate them, nor will they 610 be updated during the ATLS implementation. No sensors in buildings are planned.

## 5.2 Key findings

The key findings of the risk assessment are shown in Figure 8 and 9 and summarized below:

• The overall risk for an individual to die in a building within a radius of 2 km around the well (Figure 8) is assessed to be below $10_{-7}$ or at 0.1 micromort (1 micromort = unit of risk defined as one-in-a-million chance of death). This value is 615 within the acceptable range when compared to acceptance criteria applied in the Netherlands (or Switzerland). Reason for the acceptable risk is the estimated low vulnerability of the building stock, the overall quite limited injection volume and the fact that the initial stimulation has not produced M>2 seismicity.

• The chance of damage to buildings is around 0.1% (Figure 9) and therewith below the $10^{-2}$ acceptance threshold we have arbitrarily introduced for damage.



• The thresholds proposed in the classical traffic light (Figure 2) are consistent with the risk thresholds computed; it is not suggested to define more conservative TLS thresholds at this point.

• The uncertainties at this stage of the project are very high, highlighting the importance of updating the risk study continuously as new data becomes available.

In Appendix A1 we finally introduce a series of recommendations, that should possibly be implemented before and during the
stimulation.

**Appendix A: Recommendations**

Below, we list a number of recommendations on risk management that were made to Reykjavik Energy before the start of stimulation operations at the Geldinganes site. These are based partially on this study, but also consider experiences of past projects.

1. Excellent seismic monitoring and reliable near-real time processing is a key requirement for updating the a-priori risk assessment. The network installed at Geldinganes should be capable of this task; however, owing to the low seismicity in the region and short deployment time of the full network, the actual capabilities and operation procedures are untested.

2. The standard traffic light system operated by ISOR on behalf of OR and based on IMO magnitudes, is critically important und the ultimate decision tool. A TLS is a well-proven and well-established technology, it cannot and should not at this stage be
replaced with more adaptive concepts of risk assessment.

3. Given the uncertainties, updating this risk assessment is a key requirement. The most basic approach is to update it based on periodical re-assessment of the model parameters (seismicity rates, b-value, hydraulic parameters) performed offline and interactively. Ideally, the updating can also be performed in near-real time, and largely automated. However, this approach has never been tested under operational conditions and may fail due to unforeseen problems. ETH Zurich is aiming to make
both a periodic re-assessment as well as an automated re-assessment available to support OR, GFZ and ISOR throughout the stimulation phase in the decision taking. A detailed description of these ATLS decision support will be provided shortly before the stimulation. In addition to the risk assessment, we will also provide basic hydraulic properties of the reservoir that can help to judge the performance of the stimulation and steer further developments.

4. A pre-stimulation test that results in a number of micro-earthquakes below Magnitude 1, followed by a subsequent update of
this risk study would help to calibrate the seismic forecast models and to allow to constrain the uncertainties. The test would also demonstrate the ability of the monitoring network to detect and locate microseismicity. However, designing such a pre-stimulation phase at Geldinganes area is a difficult task. In fact, despite circa 5,000 m³ of water were injected (at low pressures)





at beginning of 2019, no seismicity was detected (with a seismic network with a magnitude of completeness of circa 0.3). Moreover, the 2001 multi-day stimulation (performed at higher rates and pressures) has resulted in no seismic events detected by the 2001 Icelandic regional seismic network.

The stimulation plan consists of three distinct stages at different sections of the well. The design of this first stage is conservative, with slowly increasing flow rates (1 hr injection per-flow rate stage) and longer shut-in phases (2 hr shut-in after each flow rate increase). Moreover, the volume injected (despite the target of 6,000 m$_3$) strictly depends on the required pressures for fracture opening/shearing (i.e., fluid-induced events). (If anomalous seismic behavior is detected or complications with the seismic network are emerging, the first stage will be modified or (eventually) stopped, as indicated by the TLS protocol). As a consequence, the calibration of the seismic forecast models is performed along with the first stage. Then, the updated values are used as prior information for the second and third sub-stimulations.

5.  The project is not seismic risk free, there is a residual chance that, despite all mitigation measures applied, damaging earthquakes might occur. This report attempts to quantify this chance, and we believe it is important to openly communicate to the public and authorities this remaining risk and the steps taken to reduce and control it. This might include clarification on how potential damages would be reported, settled and insured.

6.  Re-activating pre-existing and tectonically pre-stressed larger fracture zones and eventually triggering a larger earthquake as it happened in Pohang, is unlikely, but still probably the most important risk for the project. The probabilistic risk approach applied here captures this chance to trigger such an event to a certain extent, and in a statistical approximation. However, it may possibly underestimate the chance of such a 'triggered' earthquake if an unknown major fault is very close to the injection site (i.e., closer than 1 kilometres). Moreover, in this project, a fault zone may potentially be cause for the high temperature. Observe, in fact, that in Iceland the fault zones are oftentimes the targets of geothermal wells. Therefore, we suggest that the seismicity analyst team should be on the look-out for lineament potentially indicative of a major fault zone being re-activated, and discuss it with the experts group that is accompanying the project. In particular, a in depth analysis should be carried out after the first stage of the stimulation.

7.  The size distribution of induced earthquakes critically determines the risk, and an unusually low $b$-value may indicate the presence of critically stressed faults and will result in much larger probability of larger events. A low b-value at the same time will result in lower number of small events, which might be misinterpreted as a re-assuming low seismogenic Index. The re-assessed $b$-value must flow into the update of the risk assessment, but we suggest adding as an additional safety criterion a project halt if the $b$-value of induced events is estimated below 0.8.



8.  It is universally accepted that the seismicity and thus risk will decrease once the injection has been stopped. It is less clear, however, if gradual pressure reduction, shut-in, bleed off, or actively pumping out (if possible) are the best mitigation strategies. In this project, a common agreement has been reached on considering bleed off as the most adequate strategy.ß

9.  Surprising developments are possible, if not likely. Therefore, we set up a small interdisciplinary expert group that can come
together rapidly (e.g., virtually) if unexpected developments occur (lineaments, clusters, etc.).

*Author contribution.* Marco Broccardo conceptualized and prepared the manuscript with contributions from all co-authors, and conducted the formal analysis. Mignan Arnaud helped in the conceptualization and preparation of the study and provided the data set of Table 1. Francesco Grigoli conceptualized the mitigation strategy and prepared Section 2.4. Dimitrios Karvounis
conceptualized the Source Model 2 (Section 3.1.2) and conducted the computational analysis leading to the synthetic catalog. Together with Antonio Pio Rinaldi Dimitrios Karvounis also prepared Section 2.2. Laurentiu Danciu supported the selection of the GMPEs and GMICE and contributed to the hazard analysis. Hannes Hofman supported the preparation, creation, and presentation of the study in all its aspects. Vala Hjörleifsdóttir, Claus Milkereit, Torsten Dahm, and Günter Zimmermann reviewed and improved this study. Stefan Wiemer supported the conceptualization, preparation, and presentation of the study in
all its aspects, with a specific focus on the final discussion and recommendations.

*Data availability*. The data used in this paper is available from the authors upon request.

*Competing interests*. The authors declare that they have no conflict of interest

*Acknowledgements & funding*
This paper has been supported by DESTESS, a projected which has received funding from the European Union's Horizon 2020 research and innovation programme under grant agreement No.691728. Francesco Grigoli is funded by the European
Union's Horizon 2020 research and innovation programme under the Marie Skodowska-Curie grant agreement (no. 790900).

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

**List of Tables**

**Table 1.** Underground seismic feedback to deep fluid injection.

| Site (country*, year) | $a_{fb†}$ | $b$ | $\lambda_{M \geq 2}$ | References |
|---|---|---|---|---|
| 1-Ogachi OG91 (JP, 1991) | -2.6 | 0.7 | 4.3800 | Dinske and Shapiro (2013) |
| 2-Ogachi (JP, 1993) | -3.2 | 0.8 | 0.6942 | Dinske and Shapiro (2013) |
| 3-Soultz (FR, 1993) | -2.0 | 1.4 | 0.6942 | Dinske and Shapiro (2013) |
| 4-KTB (DE, 1994) | -1.4 | 0.9 | 27.6359 | Mignan et al. (2017) |
| 5-Paradox Valley (US, 1994) | -2.4 | 1.1 | 1.1002 | Mignan et al. (2017) |
| 6-Soultz (FR, 1995) | -3.8 | 2.2 | 0.0003 | Dinske and Shapiro (2013) |
| 7-Soultz (FR, 1996) | -3.1 | 1.8 | 0.0087 | Dinske and Shapiro (2013) |
| 8-Soultz (FR, 2000) | -0.5 | 1.1 | 87.3925 | Dinske and Shapiro (2013) |
| 9-Cooper Basin (AU, 2003) | -0.9 | 0.8 | 138.5078 | Dinske and Shapiro (2013) |
| 10-Basel (CH, 2006) | 0.1 | 1.6 | 34.7916 | Mignan et al. (2017) |
| 11-KTB (DE, 2004-5) | -4.2 | 1.1 | 0.0174 | Dinske and Shapiro (2013) |
| 12-Newberry (US, 2014a) | -2.8 | 0.8 | 1.7437 | Mignan et al. (2017) |
| 13-Newberry (US, 2014b) | -1.6 | 1.0 | 11.0021 | Mignan et al. (2017) |

* ISO code; † referred to as seismogenic index in Dinske and Shapiro (2013).


**Table 2**: List of GMPEs used in this study

| GMPE name | Location | Reference |
|---|---|---|
| 1-AB10 | Europe & Middle East | Akkar Bommer (2010) |
| 2-CF08 | Worldwide | Cauzzi Faccioli (2008) |
| 3-Zh06 | Japan | Zhao *et al.* (2006) |
| 4-Am05 | Europe and Middle East | Ambraseys et al. (2005) |
| 5-DT07 | Greece | Danciu and Tselentis (2007) |
| 6-GK02 | Turkey | Gülkan and Kalkan (2002) |
| 7-RS09 | Iceland, Europe and MiddleEast | Rupakhety and Sigjörnsson (2009) |






**List of Figures**

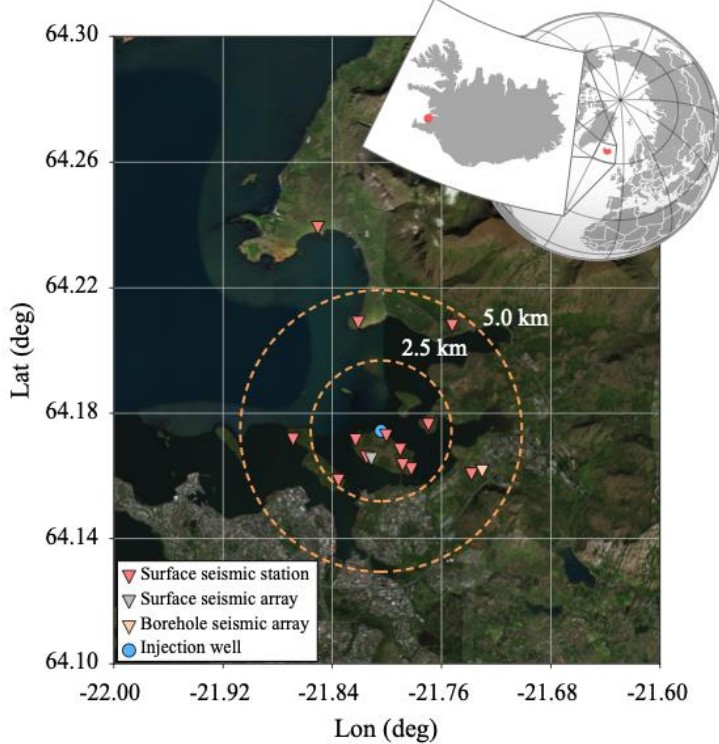

**Figure 1** Map view of the Geldinganes island, the injection well, and seismic network. Source of the map: © Google-Maps.


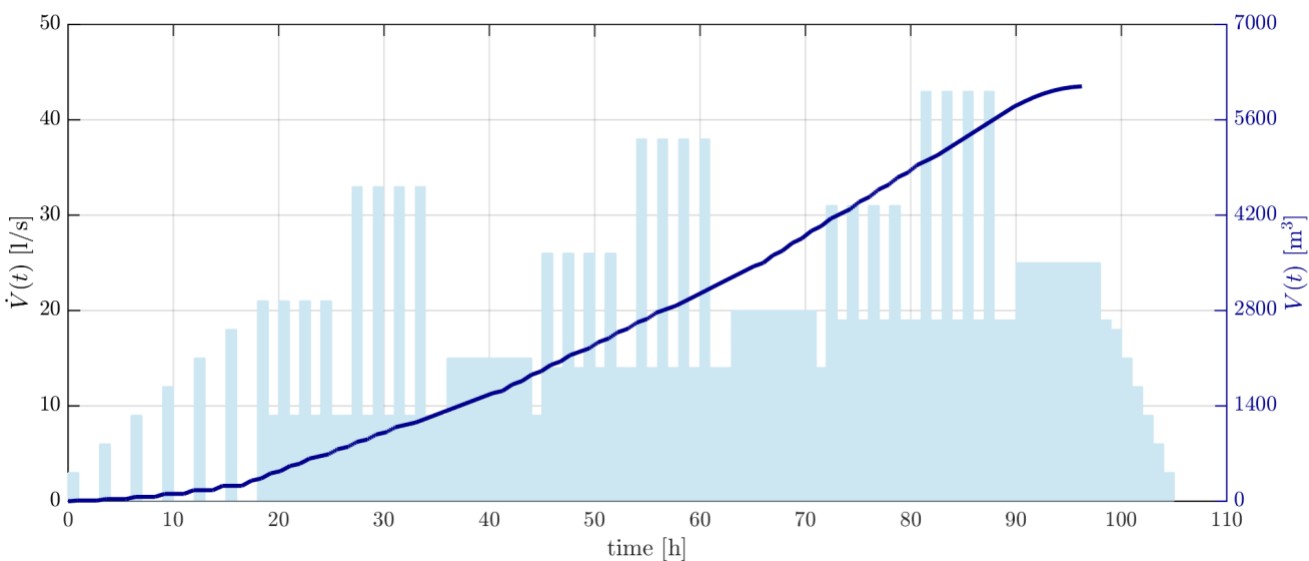

**Figure 2** Example of the main stimulation for one stage. After stimulation, flow back is performed.

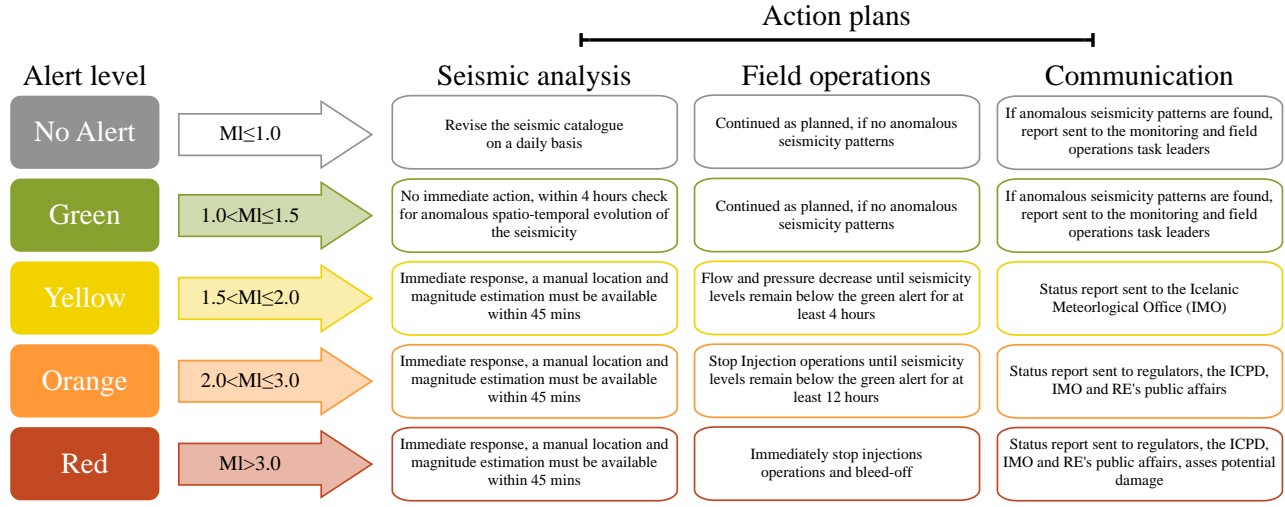

**Figure 3** The classic Traffic Light Scheme adopted in the Geldinganes project





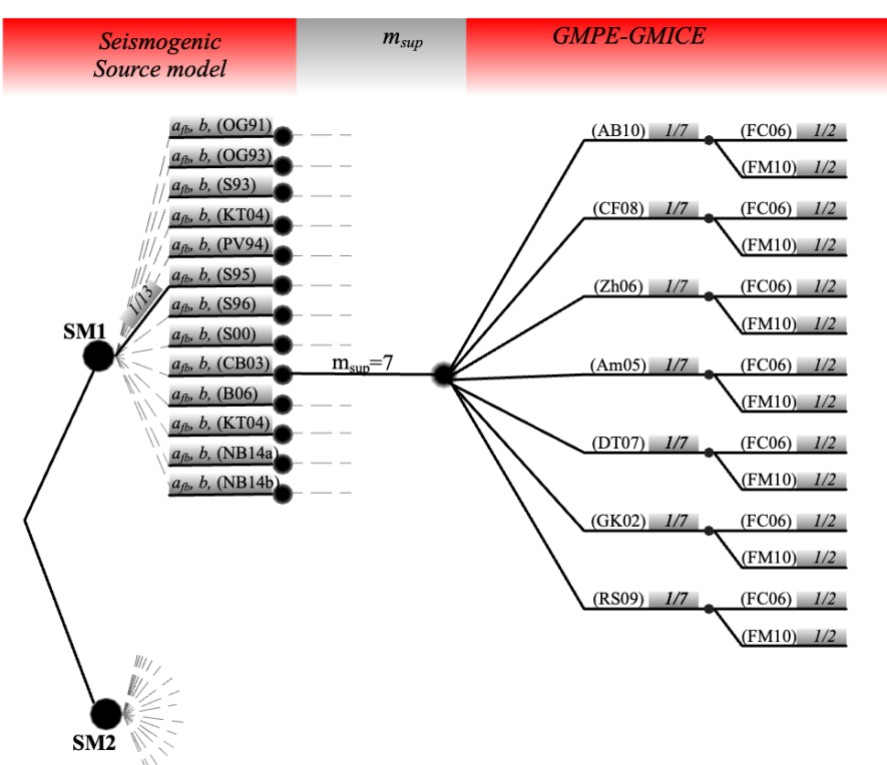

**Figure 4** Logic tree for the PSHA analysis.



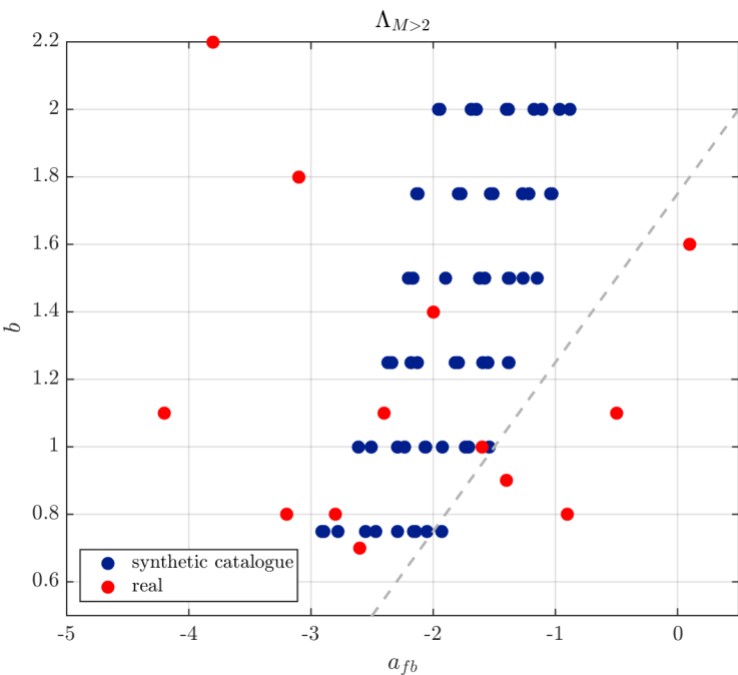

**Figure 5** Distribution of $a_{fb}$- $b$ values for the synthetic catalogue together with the dataset of Table 1. In the planned injection profile (see Figure 2), the flow rate decreases progressively back to zero, meaning that this simple model cannot strictly be applied. As approximation, we use max($\Delta V$) = 1,728 m3/day instead of $\Delta V_{shut-in}$. A direct comparison can be made between the volume injected $V$ = 18,000 $m^3$ and the equivalent $\tau\Delta V$ = 2,880 m3 for $\tau$ = 1 day and 28,800 m3 for $\tau$ = 10 days. The dashed line represents the upper limit of no expected seismicity $M > 2$.


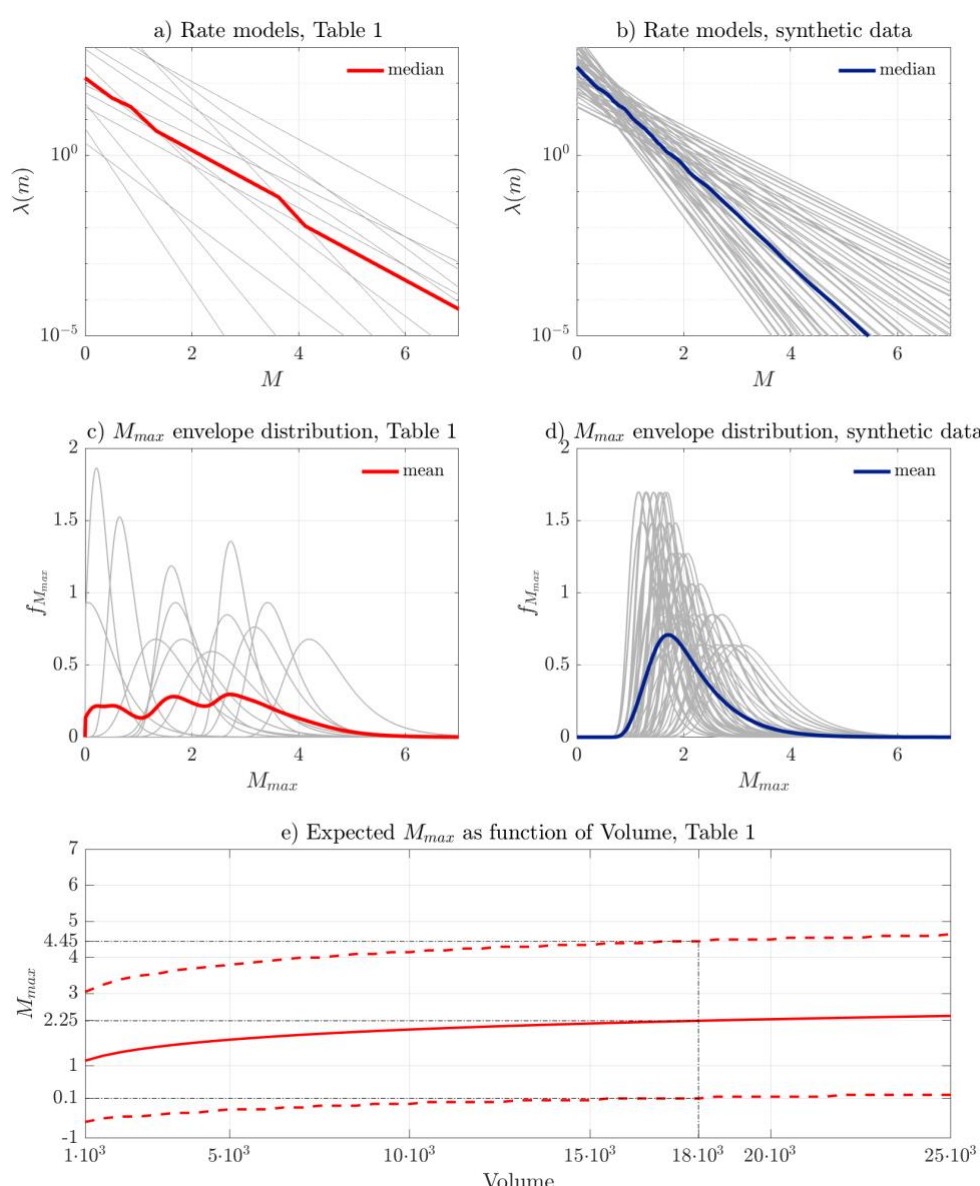

**Figure 6** Envelope probabilistic density distribution of the rate model and maximum observed magnitude $M_{max}$ a,c) based on Table 1, b,d) based on synthetic catalogue (S2 source model). e) Expected Magnitude per volume injected, based on Table 1.



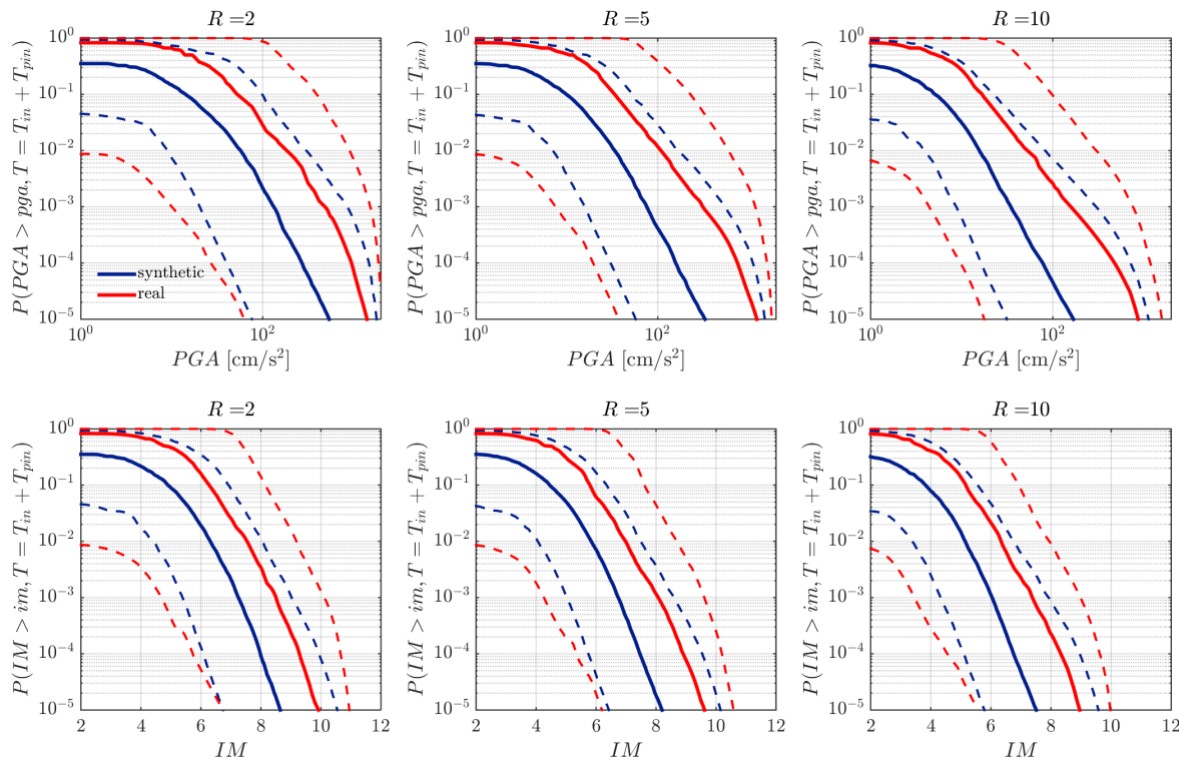

**Figure 7** PSHA analysis comparison between source model SM1 (Table 1) and SM2 (synthetic catalogue). Solid lines: medians; dashed lines 10% and 90% quantiles. Intensity measure $EMS98$.







**Figure 8** Marginal **IR** for 2 km (top) and 5 km distances based on the final model (combined SM1 and SM2). The solid horizontal lines
represent the weighted median values of the 1022 (13[$a_{fb}$, $b$]X7GMPEsX2GMICE, weight .5, + 60[$a_{fb}$, $b$]X7GMPEsX2GMICE, weight
.5) vertical gray lines. The dashed horizontal lines represent the 10 and 90% epistemic quantiles.


**Figure 9** Marginal **DR** for the final model for 2 km and 5 km distance. The solid horizontal lines represent the median values of the 511 (13[$a_{fb}, b$]X7GMPEs, .5 weight, + 60[$a_{fb}, b$]X7GMPEs, .5 weight) vertical gray lines. The dashed horizontal lines represent the 10 and 90% epistemic quantiles.






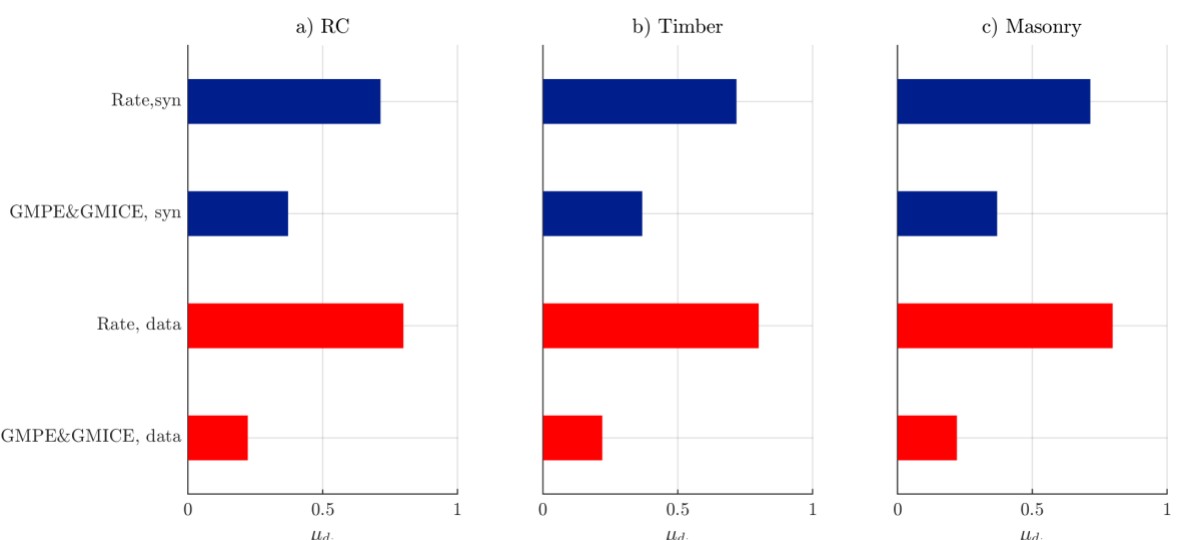

**Figure 10** Sensitivity analysis of $IR$ (observe that the QoI is $\log IR$) based on the sensitivity measure $\mu_{d_i}$ for each building class

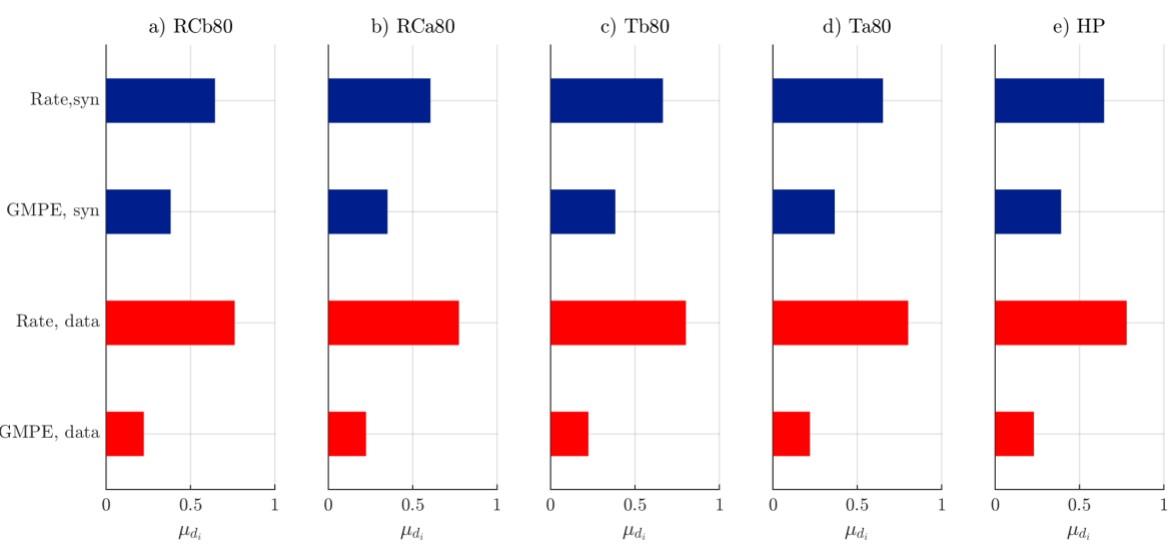

**Figure 11** Sensitivity analysis of $DR$ (observe that the QoI is $\log DR$) based on the sensitivity measure $\mu_{d_i}$ for each building class
