# Peer review of "Induced seismicity risk analysis of the hydraulic stimulation of a geothermal well on Geldinganes, Iceland"

_Natural Hazards and Earth System Sciences, 2019_

## Referee Comment (RC1) · Georgios Michas (Referee) · 27 Dec 2019

The paper presents a probabilistic assessment of the induced-seismic hazard and risk associated with a geothermal energy extraction project and the hydraulic stimulation of a deep well in the area of the Geldinganes peninsula, Iceland. Similar projects in other places around the globe have induced in the past intense microseismicity, but also larger magnitude events that have caused building damages, injuries and severe economic losses. Taking this hazard into account, the authors present a comprehensive study of the associated hazard and risk prior to the hydraulic stimulation of the well and discuss the mitigation strategies during the implementation of the project. In particular,

they estimate peak ground acceleration (PGA) and seismic intensities values based on previous induced seismicity data from similar projects and on model-produced synthetic catalogues. In addition, they estimate the seismic risk for the population and buildings, quantify the associated uncertainties and discuss possible limitations. Such studies, based on a multidisciplinary approach that combines the existing knowledge available to the scientific community, are essential for an efficient risk estimation and mitigation, as well as for the sustainability of similar projects. Overall, the paper is well-structured, well-written and presents in a comprehensive way the hazard, risk, their uncertainties and the limitations that the scientific community faces regarding hydraulic stimulation operations and fluid-induced seismicity. Therefore, I strongly recommend its publication to Natural Hazards and Earth System Sciences. However, there are a couple of points that need some clarifications. In the mitigation strategy section, the authors describe the defined alert levels according to the traffic light scheme that was planned for the project and explain the planed actions once anomalous seismicity patterns are identified (lines 203-205). While they set the magnitude thresholds for the different levels of alert and discuss b-values changes that might require further actions, further explanation is perhaps needed for the seismicity patterns in the space-time domain that might be considered "anomalous". Accordingly, while the authors recommend the use of more sophisticated techniques for operational seismologists in order to identify seismicity patterns that resemble potential faults (lines 210-211), these techniques remain vague in the text.

---

## Referee Comment (RC2) · Julian Bommer (Referee) · 13 Jan 2020

**"Induced seismicity risk analysis of the hydraulic stimulation of a geothermal well on Geldinganes, Iceland"**

**M. Broccardo, A. Mignan, F. Grigoli, D. Karvounis, A.P. Rinaldi, L. Danciu, H. Hoffmann, C. Milkereit, T. Dahm, G. Zimmermann, V. Hjörleifsdóttir & S. Wiemer**

This is an interesting paper that narrates the development of an induced seismic risk model for planned geothermal stimulations through fluid injections in Iceland. The topic would clearly seem to be within the scope of *Natural Hazards and Earth System Sciences* and, as the authors point out, it may be the first published induced seismic risk study performed prior to the commencement of injections. I would therefore conclude that the paper is a potentially valuable contribution and should be published, but I believe that it can also be improved. In particular, some information that is extraneous given the title and focus of the paper could be removed, and some more detail provided on essential features. I also think the authors could do a slightly better job of citing relevant literature and acknowledging work by others. In the following paragraphs I offer my observations and suggestions in the order in which they refer to the manuscript rather than any hierarchy of importance. I am not correcting inconsistencies in the citation of references within the text—years sometimes in brackets, sometimes not— because I do not consider that part of my work as a peer reviewer.

Line 28 (and many other locations): The Latin phrase "*a priori*" should be written in italics and never hyphenated.

Line 43: Please explain what straddle packers are since this may not be common knowledge for all readers (myself included).

Line 58: "Adding an additional" is very inelegant, please re-word.

Line 59: Heat cannot be effectively transmitted over distance so such systems will always be close to consumers out of sheer necessity.

Line 61: From the references cited here, the reader is left with the impression that other than Dr Mignan and his co-workers, nobody has ever written about the assessment and management of induced seismicity.

Line 68: "even more importantly"

Line 69: "are the sizes and stress state"

Lines 76-77: Successful? Majer et al. (2007) documents several geothermal projects in which the largest induced events occurred after pumping had stopped, highlighting the limitation of TLS. The recent paper in SRL by Baisch et al. (2019) on the efficacy of TLS is probably worth citing in this regard

Lines 81-82: It is, at least, the first being made public—there may well be commercial and confidential studies that have not been released.

Line 84: The Latin phrase "*in situ*" should be written in italics and never hyphenated.

Lines 104-105: Why or how is this being suggested? Is M 2 the detectability threshold?

Line 122: "a strike-slip/thrust regime": I am not clear if this means one or the other, or an oblique combination of the two? If the latter, do you mean reverse rather than thrust? Thrust specifically refers to very shallow dipping reverse faults, and strike-slip on such faults is unusual.

Lines 147-148: "The concept…….(Pittore et al., 2018)" – unless you tell the reader what the outcome of the DESTRESS evaluation was, this sentence does not really serve any purpose.

Line 165: Suggestion: "is planned to be approximately equal" (current wording is strange)

Line 196-236: This text is not needed since it is perfectly summarised in Figure 3 and is not central to the title and focus of the paper.

Line 238: Bommer et al. (2017) is not listed in the references—and I don't know what paper it refers to!

Line 292: "(and generally unknown *a priori*)"

Line 293: Langenbruch et al., 2018.: Nature Communications 9:3946, doi:10.1038/s41467-018-06167-4 have made the connection to pressure changes explicit in their modification of the seismogenic index.

Lines 298-299: No, it is not also known as the seismogenic index, it is the seismogenic index and you are choosing to give it another name, for reasons that are not clear to me at all. This was pioneering work by Serge Shapiro and there is no need whatsoever to assign it a new name—unless the plan henceforth is to cite this paper as the origin of the concept, which would be totally unacceptable.

Lines 314-315: What weights are assigned to each of the models? This is not indicated in Figure 4. The rationale for the weights also needs to be presented.

Lines 352-354: Again, the weights need to be reported, together with their rationale. Why are these parameter combinations not depicted as logic-tree nodes and branches in Figure 4?

Line 361: Is this really the appropriate reference for the concept of maximum magnitude?! Did nobody work on these topics before you?

Lines 363-365: This statement may not hold for induced seismicity. In some hazard and risk studies for induced earthquakes, the results are very sensitive to the maximum magnitude.

Lines 371-372: The phrase in parentheses does not currently make sense, re-word.

Lines 386-391: A clearer explanation of the two concepts could be given. Is this the difference between maximum possible and maximum expected, as discussed, for example by Zöller & Holschneider, 2016, BBSA, 106(6), 2917–2921? As currently written it will create confusion because in common practice, $M_{max}$ is used as the symbol for the parameter you refer to as $m_{sup}$.

Lines 422-423: This is a poor reason for using only PGA unless it is also found to be an efficient and sufficient parameter for the risk calculations.

Lines 423-424: Given that $R_{jb}$ ignores focal depth, which is critical for induced earthquakes (as note in the next paragraph) what was the rationale for this choice?

Lines 425-426: In Figure S1, the range of epistemic uncertainty appears to decrease with increasing magnitude, which is counter intuitive. Do the authors have an explanation for this feature?

Lines 428-429: I don't think it's probably biased, it is biased—see Bommer et al., 2007, BSSA, 97(6), 2152-2170 for discussion of this issue. Baltay & Hanks, 2014, BSSA, 104(6), 2851–2865 provide a clear physical explanation for this behaviour.

Lines 432-434: Depth influences the length of the travel path and the stress drop, so this dismissal of the issue seems rather casual.

Line 435: "over instrumental intensity measures" (physics-based is not appropriate).

Line 439: You claim to be using EMS98 but choose a GMICE calibrated in terms of MCS intensities. Musson et al., 2010, JOSE, 14, 413-428, have pointed out that while these two scales appear equivalent in their definitions, they are generally not used in an equivalent manner.

Line 440: Please explain how the increase sigma value is applied in the hazard and risk calculations—is the PGA to intensity conversion performed within the hazard integral?

Line 468: I would suggest mentioning physical damage to buildings, from which injuries and costs are both then derived.

Lines 473 & 475: What is meant by "statistically average" in these phrases? Is individual risk average over the exposed population or reported for individual locations?

Line 481: Risk targets for individuals in the Netherlands are define at $10^{-4}$ and $10^{-5}$.

Line 481: the reference is van Elk et al. (2017).

Line 495-496: On what basis is this assumption made? Is there any basis for its justification? Or you have to assume it to make the results meaningful?

Section 4: There is nothing about the exposed building stock in terms of numbers of buildings and their location, just as there is nothing about the exposed population. More information is needed and must be added.

Section 4.1: You also do not explain how hazard and risk are convolved—do you convolve hazard and fragility curves or use stochastic event sets? If the former, any spatially aggregated risk metric will be over-estimated.

Lines 565-585: Should the objectives not be stated at the beginning rather than the end?

Line 568: The concept of "informed technical community" is now considered confusing and unhelpful, and in both NUREG-2117 and NUREG-2213—which effectively supersede NUREG/CR-6372)—the expression is "centre, body and range of the technically-defensible interpretations".

Lines 603-604: Why? If there is a ground-motion recording network and large uncertainty in the choice of GMPEs, what is the basis for dismissing *a priori* any update of this part of the model?

Appendix A: I propose that you remove this section to make room for some information about the exposure model. I especially think this should come out since the recommendations only focus on modifying the model as the project proceeds, but no mention is made here on in Section 5 about how the results of the *a priori* study should be used before the study begins. Do the results provide confidence to proceed with the project? Do the results indicate that some changes could be made, such as to location or even building strength? Surely the value of a risk model is to inform decision-making? I would refer the authors to van Elk et al., 2019, Earthquake Spectra, 35(2), 537-564 for an example of an induced seismic risk model designed for such a purpose. If the value of the authors' model is to have a starting point for updating and refining during the project procedures, then this should be stated. And how are the thresholds on the TLS in Figure 3 linked to the risk estimates? Where are the disaggregation results that support these magnitude thresholds? The authors could tie the various elements of the paper together much more than is currently done.

**Julian J Bommer**
Imperial College London
13th January 2020

---

## Author Comment (AC2) · 24 Mar 2020

We thank the reviewer, Prof. Jullian Bommer, for his efforts in reviewing and improving our manuscript. We have tried to address each of his concerns in the attached document. However, some comments should be placed in the context of the revised manuscript, which will be uploaded after this discussion. In formulating our response to Professor Julian Bommers' review, we also responded indirectly to Dr Georgios Michas' review. Therefore the following document contains both answers.

Please also note the supplement to this comment:

https://www.nat-hazards-earth-syst-sci-discuss.net/nhess-2019-331/nhess-2019-331-AC2-supplement.pdf

---

## Author Comment (AC3) · 24 Mar 2020

We are very grateful for this review. The comments and concerns raised by Dr. Georgios Michas were implicitly addressed by our response to Prof. Julian Bommer's review. Therefore, we have made a unique response considering both reviewers. At the end of the attached document, you can find specific notes regarding Reviewer 1.

Please also note the supplement to this comment:
https://www.nat-hazards-earth-syst-sci-discuss.net/nhess-2019-331/nhess-2019-331-AC3-supplement.pdf

[Figure]

**Supplement:**

**Discussion on:**

**Induced seismicity risk analysis of the hydraulic stimulation of a geothermal well on Geldinganes, Iceland**

M. Broccardo, A. Mignan, F. Grigoli, D. Karvounis, A.P. Rinaldi, L. Danciu, H. Hoffmann, C. Milkereit, T. Dahm, G. Zimmermann, V. Hjörleifsdóttir & S. Wiemer

**Introduction:**

We thank the reviewers for their efforts in reviewing and providing suggestion to improve our manuscript. We greatly appreciate their work and hope that the revised version of the manuscript will be an important improvement. We would like to address each of the points raised below. We have done so in a constructive way, hoping to stimulate a fruitful discussion. Below in blue are the answers of the authors and in black the original questions of the reviewer. We proceed first with the second reviewer, Prof. Julian Bommer, because we realized that by addressing his remarks, we also indirectly answered to Reviewer 1, Dr. Georgios Michas.

**Reply to Reviewer #2 Professor Julian Bommer**

This is an interesting paper that narrates the development of an induced seismic risk model for planned geothermal stimulations through fluid injections in Iceland. The topic would clearly seem to be within the scope of Natural Hazards and Earth System Sciences and, as the authors point out, it may be the first published induced seismic risk study performed prior to the commencement of injections. I would therefore conclude that the paper is a potentially valuable contribution and should be published, but I believe that it can also be improved. In particular, some information that is extraneous given the title and focus of the paper could be removed, and some more detail provided on essential features. I also think the authors could do a slightly better job of citing relevant literature and acknowledging work by others. In the following paragraphs I offer my observations and suggestions in the order in which they refer to the manuscript rather than any hierarchy of importance. I am not correcting inconsistencies in the citation of references within the text—years sometimes in brackets, sometimes not— because I do not consider that part of my work as a peer reviewer.

Line 28 (and many other locations): The Latin phrase "a priori" should be written in italics and never hyphenated.
We thank the Reviewer for this remark. We updated the manuscript with the correct Latin form.

Line 43: Please explain what straddle packers are since this may not be common knowledge for all readers (myself included).
We thank the reviewer for pointing this out. We updated the manuscript with a short definition of a Straddle packer; i.e. "Straddle packers not only allow isolating and injecting in selected narrow zones, but also allow adjusting the straddle distance between the upper and the lower injection points."

Line 58: "Adding an additional" is very inelegant, please re-word.
We thank the Reviewer for this remark. We rephrased the sentence.

Line 59: Heat cannot be effectively transmitted over distance so such systems will always be close to consumers out of sheer necessity.

The authors acknowledge that this is often true, but not in all projects. Deep geothermal projects are often only linked to electricity production, and hot water is not necessarily used for heating. Moreover, in Iceland, there are geothermal power plants at relatively long distances (over 40 km) from Reykjavík. It is possible to transport hot water, just depends on the context and site conditions. In fact, they produce hot water at exceptionally high rates and at such high temperature that allows transmitting the heat at such distances (Mignan et al., 2019).

Line 61: From the references cited here, the reader is left with the impression that other than Dr Mignan and his co-workers, nobody has ever written about the assessment and management of induced seismicity.
We thank the Reviewer for this remark. We added the additional references relevant to this study.

Line 68: "even more importantly"
Corrected, thank you.

Line 69: "are the sizes and stress state"
Corrected, thank you.

Lines 76-77: Successful? Majer et al. (2007) documents several geothermal projects in which the largest induced events occurred after pumping had stopped, highlighting the limitation of TLS. The recent paper in SRL by Baisch et al. (2019) on the efficacy of TLS is probably worth citing in this regard
Thank you for this observation. We modified the text accordingly.

Lines 81-82: It is, at least, the first being made public—there may well be commercial and confidential studies that have not been released.
Thank you for this observation. We modified the text accordingly.

Line 84: The Latin phrase "in situ" should be written in italics and never hyphenated.
Corrected, thank you.

Lines 104-105: Why or how is this being suggested? Is M 2 the detectability threshold?
No, we did not have knowledge of the detectability threshold at that time. We considered the $M = 2$ simply as the threshold above which locals would have felt and reported an event.

Line 122: "a strike-slip/thrust regime": I am not clear if this means one or the other, or an oblique combination of the two? If the latter, do you mean reverse rather than thrust? Thrust specifically refers to very shallow dipping reverse faults, and strike-slip on such faults is unusual.
We thank the reviewer for the remark. It was indeed a mistake. We altered the text and clarify with the phrase "a strike-slip to reverse regime."

Lines 147-148: "The concept.......(Pittore et al., 2018)" – unless you tell the reader what the outcome of the DESTRESS evaluation was, this sentence does not really serve any purpose.
Thanks for the observation. After re-reading the article, we came to the conclusion that the sentence did not add any value, so we deleted it.

Line 165: Suggestion: "is planned to be approximately equal" (current wording is strange)
Thank you for the suggestion.

Line 196-236: This text is not needed since it is perfectly summarised in Figure 3 and is not central to the title and focus of the paper.
Thanks for the tip. After re-reading the article, we agreed with the reviewer and deleted the whole part.

Line 238: Bommer et al. (2017) is not listed in the references—and I don't know what paper it refers to!
Thank you for the typo detection. It is Bommer et al. 2015.

Line 292: "(and generally unknown a priori)"
Thank you for this suggestion. We modified the text accordingly.

Line 293: Langenbruch et al., 2018.: Nature Communications 9:3946, doi:10.1038/s41467- 018-06167-4 have made the connection to pressure changes explicit in their modification of the seismogenic index.
Thank you for directing us to this reference. We had missed this relationship now given between pressure and SI. We now cite it in the new version of the manuscript.

**Lines 298-299:** No, it is not also known as the seismogenic index, it is the seismogenic index and you are choosing to give it another name, for reasons that are not clear to me at all. This was pioneering work by Serge Shapiro and there is no need whatsoever to assign it a new name—unless the plan henceforth is to cite this paper as the origin of the concept, which would be totally unacceptable.
We deplore that such negative connotation seemed to transpire from our text. We have used the more generic term $a_{fb}$ (for a-value normalized by volume) since the SI infers a specific poro-elastic origin of the parameter. See for example Shapiro and Dinske (JGR 2009). This is the only reason we are reluctant to use SI terminology as the scaling can be explained by other means, including geometry (Mignan, NPG 2016) - Note however that we always mention Shapiro but we observe that using the term SI assumes, implicitly, a poro-elastic origin. It might well be the case but this remains unproven. The normalized a-value term avoids such assumption. We now clarify this aspect by: (1) highlighting the fact that Shapiro pioneered this work / was the first to find this relationship and (2) that a more generic term is used here to be agnostic regarding the underlying physics. We also added the additional Shapiro and Dinske reference. We regret that the wording may have been poorly chosen in the original version.

**Lines 314-315:** What weights are assigned to each of the models? This is not indicated in Figure 4. The rationale for the weights also needs to be presented.
Thank you for the remark. Given that "*a priori knowledge on those parameters is limited and the range of possible values wide*" we assign equal weights to the different options. We reported these weights in Figure 4 and added further explanations.

**Lines 352-354:** Again, the weights need to be reported, together with their rationale. Why are these parameter combinations not depicted as logic-tree nodes and branches in Figure 4?
Same as the previous comment

**Line 361:** Is this really the appropriate reference for the concept of maximum magnitude?! Did nobody work on these topics before you?
We added three more references on seismic risk assessment in the anthropogenic context (which also assume a truncated GR law at $m_{max}$).

**Lines 363-365:** This statement may not hold for induced seismicity. In some hazard and risk studies for induced earthquakes, the results are very sensitive to the maximum magnitude.
The reviewer is partially correct with this statement. There are indeed fluid-induced seismic studies for which the maximum possible magnitude is an important variable. These cases use as upper bound the McGarr limit (or similar), which is (usually) considerably lower than the tectonic upper bound. In our analysis, however, we use the tectonic upper bound and the output of risk analysis becomes not very sensitive to the choice of $m_{max} > 6$. See also the study of Gupta and Baker (2017). Given these considerations, we changed the text to make clear this concept.

Lines 371-372: The phrase in parentheses does not currently make sense, re-word.
Thanks, we rephrased it.

Lines 386-391: A clearer explanation of the two concepts could be given. Is this the difference between maximum possible and maximum expected, as discussed, for example by Zöller & Holschneider, 2016,

BBSA, 106(6), 2917–2921? As currently written it will create confusion because in common practice, Mmax is used as the symbol for the parameter you refer to as msup.

It is truly just a matter of notation and definitions. In the scientific literature, we can find the following (probably incomplete) list of definitions.

- Gishing and Wiemer, 2013.
  - *Maximum possible earthquake* → upper bound of the GR distribution, $M_{max}$
  - *Maximum observed earthquake* → The maximum observed earthquake during a given project, $M_{max}(obs)$
- Zöller and Holschneider 2016 & Holschneider et al. 2011
  - *Maximum possible earthquake* → upper bound of the GR distribution, $m_{max}$
  - *Maximum expected earthquake magnitude* → The maximum observed (or expected) earthquake during a given project, $M_T$
- van der Elst et al., 2016
  - *Maximum possible earthquake* → not denoted, they use unbounded GR
  - *Maximum expected earthquake magnitude* → The maximum observed earthquake during a given project, $M_{max}$

What is "common practice" depends on the context and the research group. However, in the context of fluid-induced seismicity, the van der Elst et al., 2016 seems the most popular paper (we have adopted such terminology also in Broccardo et al., 2017a).

In our paper, we tried to use consistent probabilistic terminology and notation. The rationale behind this choice is the following. First of all, the upper bound of the double truncated GR is a parameter, so it should be lowercase (as correctly used by Zöller and Holschneider). It follows that we could potentially use the notation introduced by Zöller and Holschneider, i.e. use $m_{max}$. However, in a Bayesian setting $m_{max}$, can be a random variable and therefore indicated in uppercase $M_{max}$. This $M_{max}$ creates ambiguity with the $M_{max}$ definition used in van der Elst et al., 2016 (which coincide with the $M_T$ used by Zöller and Holschneider) and also adopted by Broccardo et al., 2017a. Originally, to be consistent with this choice we have introduced for the GR upper bound $m_{sup}$, to distinguish it from the observed $M_{max}$ clearly. We wanted also to promote the idea of not using the word "maximum" for the upper bound of the GR, which will create a "probabilistic" ambiguity.

However, since we are concern about the ambiguity generated by these notations in this field, we decided to accept the reviewer's suggestions partially. Therefore, we changed notation and definitions as follow:
- New notation & definition
  - *Maximum possible earthquake (or upper bound of double truncated GR)* → $m_{max}$ (this is a parameter, lower case). Observe that the probability of observing $m_{max}$ is zero, and it cannot be "captured" by any interval.
  - *Maximum observed earthquake magnitude for a given period T* → $M_T$ (this is a random variable, uppercase). Therefore, we use the notation of Zöller and Holschneider; however, we do not use the terminology "expected maximum magnitude." Observe that this is not a very fortunate term since it will generate confusion when the expected maximum observed earthquake is of interest (i.e., $E[M_T]$).

We hope that this new notation and terminology will eliminate all ambiguities. We added also a footnote to clarify the difference between the terminology used in Zöller and Holschneider and van der Elst et al., 2016 and Broccardo et al. 2017a. Moreover, we used Holschneider et al. 2011as the reference paper.

**Lines 422-423:** This is a poor reason for using only PGA unless it is also found to be an efficient and sufficient parameter for the risk calculations.

The objective of this *a priori* risk was to collect all possible information for a first-order risk assessment of the project. (Now, we state this clearly both in the abstract and in the introduction). Therefore, only "off-the-shelf" GMPEs and fragility models were used. In addition, it is noted that local fragility

functions (used for damage risk) have been developed only for PGA (Bessason, B., and Bjarnason, 2016) using the same dataset used in the recalibration of the GMPEs. We assume that the conditions of sufficiency and efficiency and their limitations have been addressed in that study (and this is always the case for "off-the-shelf" models). Next, the fragility functions that have been used for the Individual Risk have as input the European Macroseismic Scale, which is also considered sufficient and efficient in those studies.

A more detailed bottom up approach would have required:

i) The development of an appropriate set of GMPEs for a sufficient and efficient (but also effective) IM (or a set IMs);
   a. Development of a spatial correlation model (and validation)
ii) An appropriate set of fragility for different type of buildings (which might require different IMs). This fragility analysis should have been developed by classical fragility analysis (e.g., incremental dynamic analysis) on local models and validated with an experimental campaign;
iii) A detailed microzonation (very important);
iv) A detailed exposure model and aggregate loss analysis.

However, this approach requires a consistent investment of resources and time without the possibility of implementing *a priori* a verification and validation scheme since NO local data are available.
We know that this detailed bottom-up approach has been adopted in Groningen (and other fields), *but a posteriori*, when a considerable amount of data and information (and resources) were available for validation. Observe that such an approach *a priori* is not possible without local data. In addition, allocating a considerable amount of resources for such analyses without having a first-order estimate of the hazard, based on real data, is not ideal.

Next, as we have shown in the sensitivity analysis most of the state of deep uncertainty (in this study) is due to the unknown values of $a_{fb}$ (or $\Sigma$) and $b$. So the value of information on reducing the uncertainty of these two parameters (in a first phase) is much larger than reducing the epistemic uncertainties on the GMPEs. Moreover, the (*a priori*) bias implicit in any "off-the-shelf model" is (very likely) buried under the state of deep uncertainty (and thus becomes less critical).

We are very well aware of the limitations of this study, and we have clearly listed them in Section 5. We are also aware that the following study gravitates more towards hazard rather than vulnerability and exposure; however, the first focus should be on addressing the magnitude of the hazard, i.e., if this is very low (e.g. extremely small magnitude events) it is not wise to allocate resources on having detailed GMPEs, vulnerability, and exposure models.

Therefore, we are confident that in the presence of no data, limited time, and budget constraints, the best approach is to have a first-order estimate of the hazard and associate potential risk. Then, when the project is started, and data become available, the decision-maker can demand a much more detailed analysis. Observe that while we are writing this response, the project has already terminated. In Figure R1, we report the reduction of uncertainty after a few days of the injection (a current manuscript is in preparation so no more information can be given at this level). The Figure shows the (online) projected Individual Risk. Soon it became clear that we were facing a very low-risk project (The Expected Maximum Observed Magnitude was very low). Any update of the GMPEs vulnerability and exposure would have been an excellent exercise but probably not very "valuable" for decision making.

[Figure]

*R 1* Updates of the Individual Risk and projection. Panel on the left time 0, panel on the right update after 4 days. The individual risk dropped to very low levels indicating a "very low risk" project.

**Lines 423-424:** Given that Rjb ignores focal depth, which is critical for induced earthquakes (as note in the next paragraph) what was the rationale for this choice?

It would have been ideal using $R_{hypo}$ (as we clearly state). However, we did not have such options available, and the development of low magnitude local GMPEs was not within the scope of this study (no data were available and stochastic ground motion models were not considered as an option). In fact, the development or recalibration of local GMPEs for future injections was one of the possible outcomes (in case of observable seismicity) of this experiment.

As pointed out by the reviewer, the next paragraph clearly outlines all the limitations, so that future updates can aim to eliminate them. Updates and new risk analyses can be made once the project has started, and local seismicity has been measured. We have included this strategy in the recommendations.

**Lines 425-426:** In Figure S1, the range of epistemic uncertainty appears to decrease with increasing magnitude, which is counter intuitive. Do the authors have an explanation for this feature?

This Figures are directly derived from the following paper Kowsari et al., 2019. Our conjecture is that this is due to the fact that GMPEs are recalibrated for higher magnitudes and few data are available for lower magnitude and extrapolation to lower ranges might cause a larger variability. However, the authors of the original paper are more suited to answer this question.

Lines 428-429: I don't think it's probably biased, it is biased—see Bommer et al., 2007, BSSA, 97(6), 2152-2170 for discussion of this issue. Baltay & Hanks, 2014, BSSA, 104(6), 2851– 2865 provide a clear physical explanation for this behaviour.

Thanks for these observations. We are glad that the reviewer has pointed us to these papers (we modified the text accordingly).

Lines 432-434: Depth influences the length of the travel path and the stress drop, so this dismissal of the issue seems rather casual.

See comment line 423-424 and line 422-423. The dismissal is not casual but due to the use of "off-the-shelf GMPEs." As previously mentioned, these are limitations that will (eventually) be corrected in a second phase.

Line 435: "over instrumental intensity measures" (physics-based is not appropriate).

Thanks for these observations (we modify the text accordingly).

Line 439: You claim to be using EMS98 but choose a GMICE calibrated in terms of MCS intensities. Musson et al., 2010, JOSE, 14, 413-428, have pointed out that while these two scales appear equivalent in their definitions, they are generally not used in an equivalent manner.

We thank the reviewer for this observation. However, in the context of this paper, the MSC scale is practically equivalent to the EMS98 scale. In fact, (Musson et al., 2010):

[…*In 1988 the European Seismological Commission agreed to initiate a thorough revision of the MSK Scale. The result of this work (undertaken by a large international Working Group under the chairmanship of Gottfried Grünthal, Potsdam) was published in draft form in 1993, with the final version released (after a period of testing and revision) in 1998 (Grünthal 1998). Although this new scale is more or less compatible with the old MSK Scale, the organisation of it is so different that it was renamed the European Macroseismic Scale (EMS). Since its publication it has been widely adopted inside and also outside Europe…*]

Moreover, in Faccioli and Cauzzi 2006:

[...*Based on a detailed comparison between their MSK-64 and MCS intensity ratings for all earthquakes considered, [Margottini et al., 1992] found no statistically significant difference between the two scales in the range from IV to VIII degrees (see the Appendix), in partial contrast to previous (possibly less accurate) suggestions to raise by about one degree the MCS intensity with respect to the MSK value in the range > VI [Levret and Mohammadioun, 1984]. Herein, we have extended the assumption $I_{MCS} = I_{MSK}$ also to some non-Italian earthquakes (see the Appendix)…*]

Therefore, in this study we have implicitly assumed $I_{MCS} = I_{MSK} = I_{EMS98}$. Observe that this approximation is not significant because most of the individual risk is related to moderate to significant events. We are convinced that making specific distinctions on macroseismic scales at this level of detail and with the presence of such significant uncertainties (at model rate level) is not very meaningful. Therefore, we have (just) added a small footnote to highlight this approximation.

Line 440: Please explain how the increase sigma value is applied in the hazard and risk calculations— is the PGA to intensity conversion performed within the hazard integral?

Thank you for the remark. We think we explained this quite clearly in the lines 435-450 (of the original document). We basically derived the equation for $P(IM > im|m, r)$ by converting the original GMPE into the GMICE, i.e.

$$\log(PGA) = f(M, R) + \epsilon$$

If $\epsilon$ is Gaussian (with $\sigma_{GMPE}$) then $\log(PGA)$ is Gaussian. Next,

$$I = a \log PGA + b + \varepsilon,$$

then $I$ is conditionally Gaussian with mean $aE(\log PGA) + b$ and variance $a^2\sigma_{GMPE}^2 + \sigma_{MICE}^2$.
Given that (already reported at line 440 of the original paper) we have fully specified $P(IM > im|m, r)$ and the hazard curve can be computed with the classical total probability theorem as $\Lambda(im; T, b) = -\int_m P(IM > im|M = m, r)$. Also, this was already reported at line 445 of the original manuscript. Note that in the supplement, we have also reported all the values for $a$ and $b$ for each GMICE. Given this description we did not alter the text.

Line 468: I would suggest mentioning physical damage to buildings, from which injuries and costs are both then derived.

Thank you for this remark. We amended the text accordingly.

Lines 473 & 475: What is meant by "statistically average" in these phrases? Is individual risk average over the exposed population or reported for individual locations?

It is the standard definition used for quantitative risk measures for loss of life and economic damage. See Jonkman et al., 2003. Precisely, the average person is the "expected" person within a population. In practice, one has to think about randomly selecting a person from a population (which is, by definition, the expected person and therefore the statistically average person). We have not introduced this definition, but it is formally defined in Jonkman et al. 2003 and Jones, 1992. The same reasoning can be applied to a population of different buildings. Since we gave the proper references, we believe it is not necessary to modify the text.

Line 481: Risk targets for individuals in the Netherlands are define at $10^{-4}$ and $10^{-5}$.
Line 481: the reference is van Elk et al. (2017).

Thank you very much, we corrected the text.

Line 495-496: On what basis is this assumption made? Is there any basis for its justification? Or you have to assume it to make the results meaningful?

Thank you for the remark. It is an implicit assumption (stemming from Mignan et al. 2015) since we use "off-the-shelf" macroseismic vulnerability model. In principle, it is not even necessary for this study (without it, the results are not changing). Therefore, we have decided to eliminate it.

Section 4: There is nothing about the exposed building stock in terms of numbers of buildings and their location, just as there is nothing about the exposed population. More information is needed and must be added.

Observe that at this level of details we expressly choose to not use *any* aggregate risk measures (including losses). In fact, we wrote at lines 469-471 of the original document:

[...*The a-priori risk analysis for the Geldinganes project here focuses on the first risk, while the aggregate economic losses are not directly computed. Here, as a substitute for aggregate losses, we define a low damage threshold for statistical average classes of Icelandic building...*]

As stated, we limited our analysis only on marginal risk measures. For this reason, we introduced the Damage Risk, which represents the risk of light damage given a class of buildings (regardless of the number of buildings). It is the same definition of Individual Risk but in the damage domain. This facilitates the decision-making process, making individual risk and damage risk two compatible measures. Observe that also the Individual Risk is given as a function of the building class. The *modus operandi* is the following,

- Given a location of the built environment at a distance $R$ from the injection point, we consider any information about the specific class of the built environment as missing (detailed analysis was out of the scope of the current study).
- We compute the *IR* ad *DR* as function of the Icelandic building classes.
- Therefore, for each built environment coordinate, we computed both IR and DR for a given class of buildings. Next, the decision making process is performed by defining a threshold on the worst-case scenario (i.e., the weakest of the building class).

We have explained these steps in detail throughout Section 4. Given this, the detailed information of the exposure model is not significant for the calculation and is only useful for general information. Therefore, we have included general information on the first order exposure estimation in the supplement (this information is already available in the original manuscript). Please note that we have now also included additional details that has been used in the internal risk analysis report.

Section 4.1: You also do not explain how hazard and risk are convolved—do you convolve hazard and fragility curves or use stochastic event sets? If the former, any spatially aggregated risk metric will be over-estimated.

It is explained at the end of previous section (we slight modified the text to be more precise)

[...*The framework used for the computation of IR and DR is based on the convolution of the hazard model with the vulnerability models for the relevant building types, and (only for the IR) with the consequence model...*]

As indicated in the previous comment (and clearly stated in the text), we decided not to use aggregate measures but only marginal measures (which do not depend on any correlation structure). In principle, to obtain aggregate measurements, we need a spatial correlation model based on further model assumptions (i.e., Gaussian random field) and on assumptions on the parameters of the model itself. Since no data are available, these assumptions cannot be validated. Alternatively, we can look for an upper limit (as implicitly indicated by the author) considering individual losses as comonotonic random variables, and a lower limit considering losses as statistically independent (Broccardo et al. 2017b). However, in the presence of such deep uncertainties (at rate level), these limits are essentially meaningless (i.e., they cover pretty much the entire domain of losses for any return period of interest). Such a level of detail, at a very early stage of a new project, is not recommended. We consider it essential first of all to collect data from a pre-stimulation or a first phase of the project to have the correct Hazard model and then to move (eventually) to aggregate measurements. In addition, at this stage, we also choose not to perform the decision-making process at the monetary loss level because there is large uncertainty in defining the repair and replacement costs of an individual or group of local buildings (and how these costs are related to the damage). As we have said above (now the project is finished), it was soon evident that this kind of detailed analysis was not necessary for this project.

Lines 565-585: Should the objectives not be stated at the beginning rather than the end?
Thanks a lot for the comment, we have restructured the introduction.

Line 568: The concept of "informed technical community" is now considered confusing and unhelpful, and in both NUREG-2117 and NUREG-2213—which effectively supersede NUREG/CR-6372)—the expression is "centre, body and range of the technically-defensible interpretations".
In restructuring the introduction, we have eliminated this part.

Lines 603-604: Why? If there is a ground-motion recording network and large uncertainty in the choice of GMPEs, what is the basis for dismissing a priori any update of this part of the model?
This was not completely dismissed. We did not perform online updates of the coefficients of the GMPEs because, from the sensitivity analysis, most of the uncertainty was related to the rate model. However, we did plan to updates the GMPE models itself in case sufficient seismic data were available (but this is an *a posteriori* analysis).

Appendix A: I propose that you remove this section to make room for some information about the exposure model. I especially think this should come out since the recommendations only focus on modifying the model as the project proceeds, but no mention is made here on in Section 5 about how the results of the a priori study should be used before the study begins. Do the results provide confidence to proceed with the project?
Thanks for the suggestions. We have already answered in the previous comment on the exposure model (and all extra information is in the supplement). This study represents the synthesis of a collective effort that has produced a risk report. The result (as we stated in both the abstract and the key findings in Section 5) indicated that the median of individual risk and injury risk were both below the selected threshold. As a result, the project started (now reported in Section 5). However, we clearly indicated in the abstract, introduction and limitation (Section 5) that the results are associated with a state of deep

uncertainty and several limitations. We also indicated the steps for the online update of the uncertainty of the rate model (which is a subject of a future study), and the improvements to be made if the project turns out to be in a "risky" domain. All this is now highlighted in (re-structured) Section 5. Moreover, we believe that the recommendations section is actually very important in this study. We shorten it and moved into Section 5.

Do the results indicate that some changes could be made, such as to location or even building strength? Surely the value of a risk model is to inform decision-making? I would refer the authors to van Elk et al., 2019, Earthquake Spectra, 35(2), 537-564 for an example of an induced seismic risk model designed for such a purpose

As mentioned above, this assessment would have been carried out after the start of the project if the level of risk was close to the thresholds set. In the absence of data and in the presence of such a level of uncertainty at the rate level (highlighted in the sensitivity section), providing such suggestions may not be advisable. At the present time, the project is finished. As can be seen from Figure R1, it was clear, early enough, that any suggestions for improving the strength of the building would be unnecessary. Also, changing the location of the well was simply not possible because the project is a re-stimulation of an existing well.

. If the value of the authors' model is to have a starting point for updating and refining during the project procedures, then this should be stated.

We believed that this was clear since the abstract. In fact we wrote:

*[…However, these results are affected by several orders of magnitude of variability due to the deep uncertainties present at all levels of the analysis, indicating a critical need in updating this risk assessment with in situ data collected during the stimulation…]*

However, since it was not clear enough, we modify and stress the true purpose of the study in both abstract, introduction, and conclusion.

And how are the thresholds on the TLS in Figure 3 linked to the risk estimates? Where are the disaggregation results that support these magnitude thresholds? The authors could tie the various elements of the paper together much more than is currently done.

Thank you for that question. The thresholds for the classic TLS were chosen following previous projects. In Iceland the classic TLS has already been applied for example for re-injection activities in the geothermal field of Hellisheidi, a region not far from Reykjavik (Thorsteinsson and Gunnarsson, 2014). However, for additional precautions (since we were closer to the city), we reduced each alert level by 0.5 units of magnitude. In addition, since our seismic network allows to analyze events with $Ml<1.0$ we have added an additional stage (no alert level) in which routine seismologists can evaluate in an advanced way the space-time evolution of induced seismicity using advanced seismic analysis tools. Observe that the Traditional TLS has been set separately from this study. While the Advance Traffic Light System makes use of this a priori risk analysis (and it is an experiment).

Thorsteinsson, H. and Gunnarsson, G.: Induced Seismicity – Stakeholder Engagement in Iceland, GRC Transactions, 38, (2014), 879-881

**Reply to Reviewer #1 Dr. Georgios Michas**

The paper presents a probabilistic assessment of the induced-seismic hazard and risk associated with a geothermal energy extraction project and the hydraulic stimulation of a deep well in the area of the Geldinganes peninsula, Iceland. Similar projects in other places around the globe have induced in the past intense microseismicity, but also larger magnitude events that have caused building damages, injuries and severe economic losses. Taking this hazard into account, the authors present a comprehensive study of the associated hazard and risk prior to the hydraulic stimulation of the well and discuss the mitigation strategies during the implementation of the project. In particular, they estimate peak ground acceleration (PGA) and seismic intensities values based on previous induced seismicity data from similar projects and on model-produced synthetic catalogues. In addition, they estimate the seismic risk for the population and buildings, quantify the associated uncertainties and discuss possible limitations. Such studies, based on a multidisciplinary approach that combines the existing knowledge available to the scientific community, are essential for an efficient risk estimation and mitigation, as well as for the sustainability of similar projects. Overall, the paper is wellstructured, well-written and presents in a comprehensive way the hazard, risk, their uncertainties and the limitations that the scientific community faces regarding hydraulic stimulation operations and fluid-induced seismicity. Therefore, I strongly recommend its publication to Natural Hazards and Earth System Sciences. However, there are a couple of points that need some clarifications. In the mitigation strategy section, the authors describe the defined alert levels according to the traffic light scheme that was planned for the project and explain the planed actions once anomalous seismicity patterns are identified (lines 203-205). While they set the magnitude thresholds for the different levels of alert and discuss b-values changes that might require further actions, further explanation is perhaps needed for the seismicity patterns in the spacetime domain that might be considered "anomalous". Accordingly, while the authors recommend the use of more sophisticated techniques for operational seismologists in order to identify seismicity patterns that resemble potential faults (lines 210-211), these techniques remain vague in the text.

We are very grateful for this review. The detailed section on the TLS recommendations has been deleted following a suggestion of Reviewer #2. However, we do not want to elude the questions. For the anomalous seismic patterns, we mean a drop of the $b$ value and/or an increase of $a_{fb}$ (this corresponds to the lower left domain of Figure 5). Moreover, with "more sophisticated techniques" we intended the Relative relocation methods (Waldhauser and Ellsworth, 2000). Again, this is not included in the current text.

Waldhauser, F., & Ellsworth, W. L. (2000). A double-difference earthquake location algorithm: Method and application to the northern Hayward fault, California. Bulletin of the Seismological Society of America, 90(6), 1353-1368.